# Assigning function to natural allelic variation via dynamic modeling of gene network induction

Magali Richard[1,2,*] ID, Florent Chuffart[1], Hélène Duplus-Bottin[1], Fanny Pouyet[1], Martin Spichty[1], Etienne Fulcrand[1], Marianne Entrevan[1], Audrey Barthelaix[1], Michael Springer[3], Daniel Jost[2,**] ID & Gaël Yvert[1,***] ID

## Abstract

More and more natural DNA variants are being linked to physiological traits. Yet, understanding what differences they make on molecular regulations remains challenging. Important properties of gene regulatory networks can be captured by computational models. If model parameters can be "personalized" according to the genotype, their variation may then reveal how DNA variants operate in the network. Here, we combined experiments and computations to visualize natural alleles of the yeast *GAL3* gene in a space of model parameters describing the galactose response network. Alleles altering the activation of Gal3p by galactose were discriminated from those affecting its activity (production/degradation or efficiency of the activated protein). The approach allowed us to correctly predict that a non-synonymous SNP would change the binding affinity of Gal3p with the Gal80p transcriptional repressor. Our results illustrate how personalizing gene regulatory models can be used for the mechanistic interpretation of genetic variants.

**Keywords** galactose; personalized medicine; SNP function; stochastic model; yeast
**Subject Categories** Methods & Resources; Network Biology; Quantitative Biology & Dynamical Systems
**Mol Syst Biol.** (2018) 14: e7803

## Introduction

In the past decade, countless DNA variants have been associated with physiological traits. A major challenge now is to understand how they operate at the molecular level. This is a difficult task because the mechanistic consequences resulting from each variant are not easy to identify. Even when the function of a gene is well documented, investigators need to determine the tissues, cells, or organelles in which a mutant allele makes a biological difference, the developmental stage at which this may happen, the metabolic or regulatory network that may be involved, as well as possible molecular scenarios. A mutation may alter the regulation of transcription or mRNA splicing; the enzymatic activity of the target protein; its rate of production, maturation, or degradation; its intracellular localization; its binding affinity to an interacting partner or the specificity of its molecular interactions. In the vast majority of cases, information from the DNA sequence alone is not sufficient to delimit the perimeter of possible implications.

Systems biology has opened new opportunities to better predict the action of DNA variants. First, "omics" data that are gathered at various levels (DNA, transcripts, proteins, metabolites, etc.) establish relations between target sequences and functional pathways. Information about molecular and genetic interactions, expression profiles, chromatin landscapes, post-transcriptional and post-translational regulations can be exploited to derive functional predictions of DNA variants. Various methods have been proposed to do this, such as Bayesian genetic mapping (Gaffney *et al*, 2012), visualization of SNPs on relational protein networks (Bauer-Mehren *et al*, 2009), prioritization based on negative selection (Levenstien & Klein, 2011), inference of miRNA:RNA binding defects (Coronnello *et al*, 2012), or combinations of lncRNA eQTL-mapping with DnaseI-hypersensitivity maps (Guo *et al*, 2016). In addition, structural data of biomolecules can also highlight functional perturbations in specific domains such as catalytic sites or interaction surfaces (Barenboim *et al*, 2008; Al-Numair & Martin, 2013).

Another alternative is to model the quantitative and dynamic properties of molecular reactions and to explore which feature(s) may be affected by a DNA variant. The functional consequences of mutations can then be inferred by considering their impact on specific parameters of the model. In other words, assigning function to a DNA variant may be straightforward after it is linked to parameters of a model. This perspective may also, on the long term, generate developments in personalized medicine: If a model can be

1 Laboratoire de Biologie et de Modélisation de la Cellule, Ecole Normale Supérieure de Lyon, CNRS, Université Lyon 1, Université de Lyon, Lyon, France
2 Univ. Grenoble Alpes, CNRS, CHU Grenoble Alpes, Grenoble INP, TIMC-IMAG, Grenoble, France
3 Department of Systems Biology, Harvard Medical School, Boston, MA, USA
*Corresponding author. Tel: +33 4 56 52 00 68; E-mail: magali.richard@univ-grenoble-alpes.fr
**Corresponding author. Tel: +33 4 56 52 00 69; E-mail: daniel.jost@univ-grenoble-alpes.fr
***Corresponding author. Tel: +33 4 72 72 80 00; E-mail: gael.yvert@ens-lyon.fr

*personalized* according to the patient's genotype, then it can help predict disease progress or treatment outcome and therefore adapt medical care to the patient's specificities. Such an approach nonetheless differs from machine-learning techniques, which can be efficient for prediction but where parameters are often not interpretable. For it to become reality, the model must be (i) informative on the biological trait of interest and (ii) identifiable (variation of one parameter cannot be exactly compensated by variation of another parameter) and sufficiently constrained (few parameters with limited degrees of freedom) so that fitted parameter values can inform on the patient's specificities. These two requirements antagonize each other regarding the complexity of the model to be used. The former asks for completeness: The molecular control of the trait must be correctly covered by the model, describing known reactions as best as possible. The latter asks for simplicity: If too many parameters are allowed to be adjusted to the data, then the validity of the personalized model is questionable and none of the adjustments are informative. It is therefore important to determine if and how personalizing model parameters can be productive.

For a given molecular network, individuals from natural populations have different genotypes at several nodes (genes) of the network, as well as in numerous external factors that can affect network properties. Such external factors can modify, for example, global translation efficiencies, metabolic states, or pathways that cross-talk with the network of interest. Adapting model parameters to specific individuals is challenging when so many sources of variation exist. A way to circumvent this difficulty is to study the network experimentally in the context of a more reduced and focused variation. If investigators have access to nearly isogenic individuals that differ only at specific genes of the network, they can then characterize the differences in network behavior that result from these specific allelic differences. The numerous external factors affecting the network can then be ignored or drastically simplified in the model because they are common to all individuals. This way, the parameter space is constrained and only potentially informative parameters are allowed to be adjusted to fit individual-specific data.

Some model organisms such as the yeast *Saccharomyces cerevisiae* offer this possibility. They can be manipulated to generate single allelic changes, which provides an ideal framework to link DNA variants to model parameters. In particular, the gene regulatory network controlling the yeast response to galactose (GAL network) is well characterized, both *in vivo* and *in silico*. This circuit controls galactose utilization by upregulating the expression of regulatory and metabolic genes in response to extracellular galactose (Sellick *et al*, 2008). Regulation is based on the transcriptional activator Gal4p, the galactose transporter Gal2p, a signal transducer Gal3p, and the transcriptional inhibitor Gal80p. In addition, the galactokinase Gal1p involved in galactose metabolism is also a co-inducer of the response (Bhat & Hopper, 1992). This system can display either a gradual induction (where the rate of transcription progressively increases in each cell according to the timing and intensity of the stimulus) or a binary induction (where some cells are rapidly activated and others not). This dual behavior has received a lot of attention, and important molecular features have been elucidated by experimental and theoretical approaches (Biggar & Crabtree, 2001; Hawkins & Smolke, 2006; Song *et al*, 2010; Apostu & Mackey, 2012). In particular, the

dynamic response of a population of cells to galactose can be described by two quantities: (i) The inducibility of the network is defined as the proportion of activated cells in the population and (ii) the amplitude of the response refers to the expression level that is reached by induced cells. Regulatory feedback loops of the network are critical to the switch-like behavior. They were shown to feed back the dynamics of transcription bursts rather than the levels of expression (Hsu *et al*, 2012). They regulate the amplitude response by reducing noise in GAL gene expression (Ramsey *et al*, 2006), they control inducibility by fine-tuning the timing of the switch (Ramsey *et al*, 2006), and they participate to the memory of previous inductions (Acar *et al*, 2005; Kundu & Peterson, 2010). As a consequence, bimodal distributions of expression of the GAL genes can be observed in isogenic populations exposed to intermediate concentrations of inducer (Becskei *et al*, 2001; Venturelli *et al*, 2012; Peng *et al*, 2015), and this population heterogeneity can confer a growth advantage during the transition from glucose to galactose metabolism (diauxic shift) (Venturelli *et al*, 2015). Interestingly, wild yeast isolates present diverse types of induction dynamics during the diauxic shift, ranging from strictly unimodal to transient bimodal distribution of expression levels (New *et al*, 2014; Wang *et al*, 2015). This indicates that natural genetic variation can modify the network dynamics.

The *GAL3* gene plays a central role in the network. Its protein product Gal3p is activated by binding to galactose and ATP and then binds as a dimer to Gal80p dimers to release the repression on Gal4p at target promoters (Sellick *et al*, 2008). The protein is enriched in the cytoplasm prior to stimulation and in the nucleus after the stimulation, although this cytonuclear transfer does not account for the dynamics of activation (Jiang *et al*, 2009; Egriboz *et al*, 2011). Expression of *GAL3* is itself under Gal4p/Gal80p control (positive feedback). In addition, the sequence of *GAL3* differs between natural isolates of *S. cerevisiae* and this allelic variation was recently associated with different sensitivities of the network to galactose (Lee *et al*, 2017). There are multiple ways that a *GAL3* variant could affect the dynamics of induction: by modifying the production or degradation rates of the Gal3p protein or of its messenger RNA, by changing the affinity of Gal3p to galactose or ATP, by changing the capacity of Gal3p to dimerize, by changing the nucleocytoplasmic ratio of Gal3p molecules, or by changing the affinity of Gal3p to Gal80p. A *GAL3* variant may also affect the background expression level of Gal3p prior to stimulation, which is known to be critical for network memory of prior stimulations (Stockwell & Rifkin, 2017). Thus, it is difficult to predict the functional consequence of sequence variation in *GAL3*.

Using the yeast *GAL3* gene as a model framework, we show here that experimental acquisitions combined with network modeling are efficient to predict the effect of sequence variants. The principle of the approach is to link genetic variation to informative changes of parameter values of the model. We show that replacing natural *GAL3* alleles can be sufficient to transform a gradual response into a binary activation, and the approach allowed us to distinguish between different types of *GAL3* alleles segregating in *S. cerevisiae* populations: those altering the activation of Gal3p by galactose and those altering the strength with which activated Gal3p alleviates the transcriptional inhibition operated by Gal80p. In particular, our approach was efficient to associate a non-synonymous SNP with a change of binding affinity for Gal80p.

# Results

## Natural variation in GAL3 is sufficient to convert a gradual induction into a binary switch

We constructed a panel of yeast strains that were all isogenic to the reference laboratory strain BY, except for *GAL3*. At this locus, each strain carried an allele that was transferred from a natural strain of the *Saccharomyces* Genome Resequencing Project (Liti *et al*, 2009; Appendix Fig S1). All strains of the panel also harbored a $P_{GAL1}$-*GFP* reporter of network activity, where the promoter of the *GAL1* gene controlled the expression of a GFP fluorescent protein destabilized by a degradation signal (Mateus & Avery, 2000; Chuffart *et al*, 2016). *GAL1* is a paralogous gene of *GAL3* (Hittinger & Carroll, 2007) and transcription at its promoter is commonly used as a proxy of GAL network activity (Acar *et al*, 2005; Venturelli *et al*, 2015; Wang *et al*,

2015). Using flow cytometry, we monitored the dynamics of network activation in each strain (Fig 1). This was done by first culturing cells for 3 h in a medium containing 2% raffinose, a sugar known to be neutral on network activity, adding galactose (0.5% final concentration), and quantifying fluorescence at multiple time points for 4 h. Significant differences in the dynamics of activation were observed between the strains. Those harboring the $GAL3^{NCYC361}$, $GAL3^{K11}$, $GAL3^{BY}$, $GAL3^{DBVPG1788}$, $GAL3^{DBVPG1853}$, and $GAL3^{JAY291}$ alleles displayed a gradual response and all cells of the population were induced and responded with similar rate of expression, maintaining population homogeneity (see example shown in Fig 1A). In contrast, strains harboring the $GAL3^{Y12}$ and $GAL3^{YJM978}$ alleles displayed a binary response, with a transient coexistence of induced (ON) and uninduced (OFF) cells in the population (example in Fig 1B).

We quantified induction using two metrics: the mean level of reporter expression in activated cells (response amplitude) and the

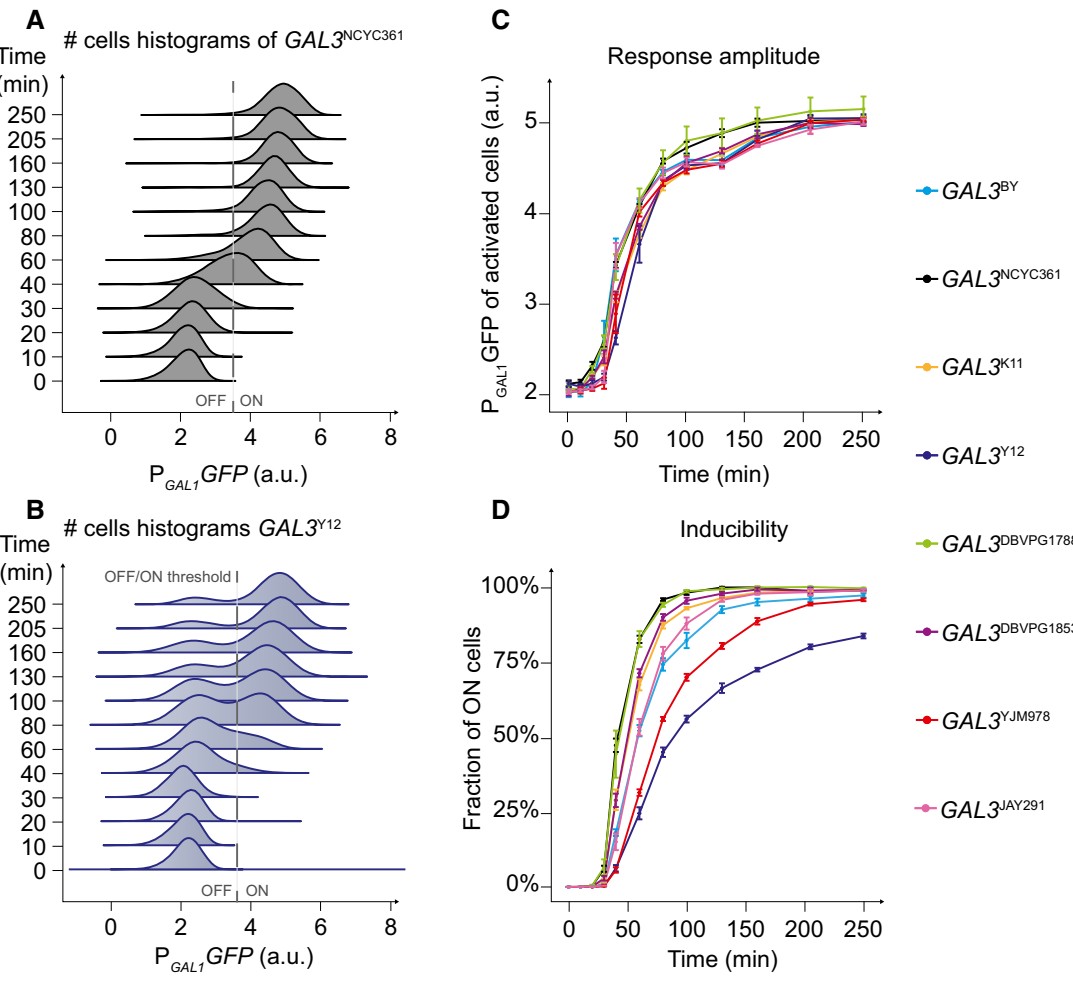

**Figure 1.  Dynamic response to galactose in the context of *GAL3* variants.**

Acquisitions were made on strains where the *GAL3* allele was replaced by the indicated natural alleles. These strains were otherwise isogenic, with a BY background.

A, B     Flow cytometry data obtained on strains harboring the $GAL3^{NCYC361}$ allele (A) or the $GAL3^{Y12}$ allele (B). Cells were cultured in raffinose 2% and induced at time 0 by adding galactose at a final concentration of 0.5%. a.u., arbitrary units. Gray dashed line, threshold used to distinguish ON cells from OFF cells.

C      Amplitude of the response (mean expression) as a function of time for each *GAL3* replacements strain. Error bars represent standard error of the mean (*n* = 6).

D      Inducibility of the response (fraction of ON cells) as a function of time for each *GAL3* replacement strain. Error bars represent standard error of the mean (*n* = 6).

proportion of activated cells in the population (inducibility of the network). We observed that the response amplitude varied little among the strains, all of them approaching steady state with comparable kinetics (Fig 1C). In contrast, inducibility of the network differed between strains (Fig 1D). As expected, in strains showing a gradual response, the fraction of ON cells increased significantly during the first 2 h of induction, reaching full inducibility (all cells activated) by the end of the experiment. On the opposite, the strains showing a transient binary response displayed reduced inducibility over time. For instance, 21% of *GAL3*[Y12] cells were still not induced after 250 min of stimulation. These results indicate that natural genetic variation in *GAL3* is sufficient to modify the inducibility of the network and to convert a gradual response into a binary response, or *vice versa*.

## A quantitative model of inducibility over time

To examine what functional properties of the *GAL3* gene could determine a gradual or a binary response, we constructed a dynamic stochastic model of the network (Fig 2A). We based our quantitative model on the following current molecular knowledge, which derives from reference laboratory strains. In absence of galactose, a homodimer of the transcription factor Gal4p is constitutively bound to upstream activation sites (UAS) of promoter regions of GAL genes. However, transcription is inactive because of the homodimeric Gal80p inhibition of Gal4p (Peng & Hopper, 2002; Pilauri *et al*, 2005). When intracellular galactose binds Gal3p, it changes conformation and associates with Gal80p dimers (Lavy *et al*, 2012), thereby releasing Gal80p from promoters and allowing

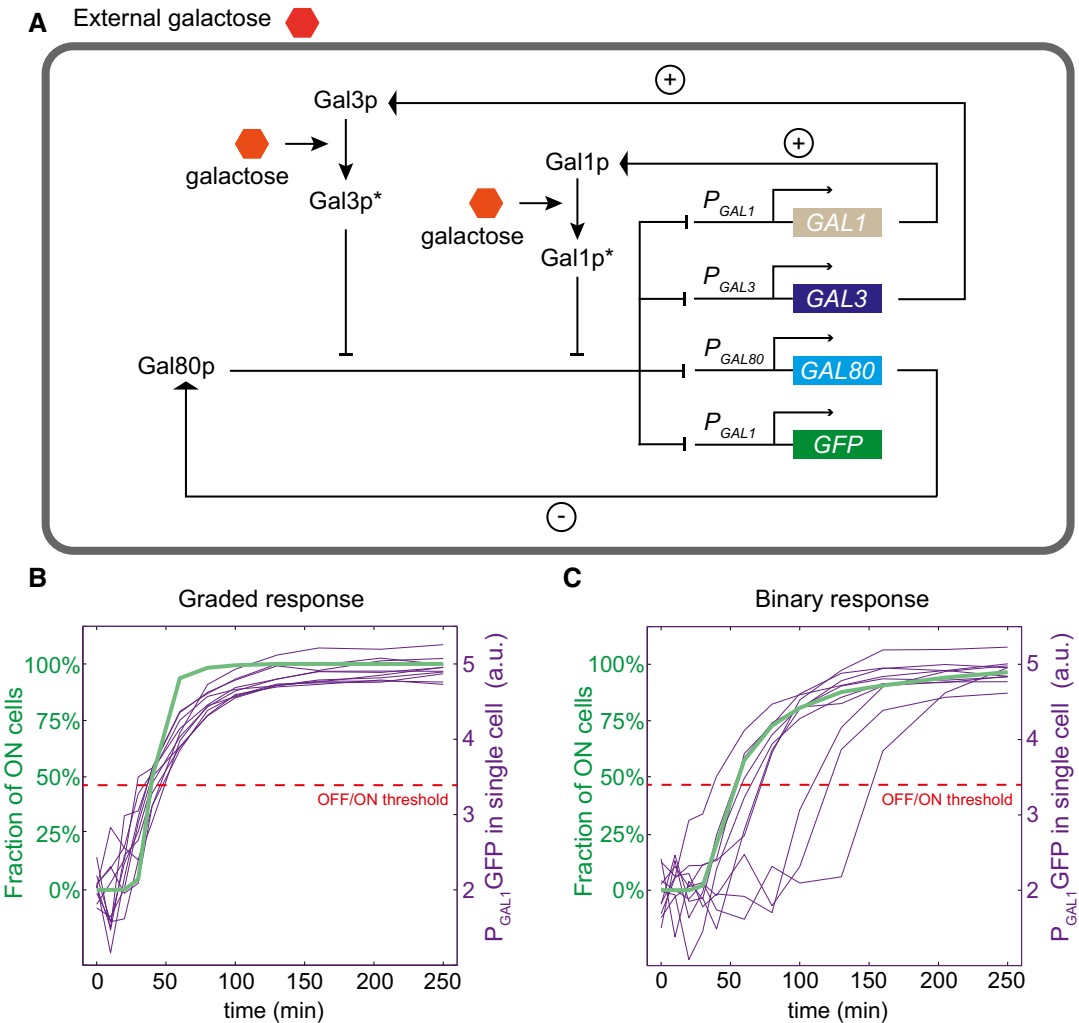

**Figure 2.** *In silico* model of network induction.

A   Schematic representation of the model used. Galactose-activated Gal1p and Gal3p proteins become Gal1p* and Gal3p*, respectively. Pointed and blunt arrows represent activation and inhibition, respectively. Positive and negative feedback loops are highlighted by + and − signs. The central Gal4p activator is not shown because its dynamics is not included in the model.

B   Example of a gradual response predicted by the model ([gal] = 0.5%, $\rho_{Gal3}$ = 140 and $K_{Gal}$ = 0.055). Thin violet lines represent stochastic simulations of network activation in individual cells. Dashed red line represents the threshold distinguishing ON from OFF cells. Green thick line indicates the fraction of ON cells as a function time.

C   Example of a binary response predicted by the model ([gal] = 0.5%, $\rho_{Gal3}$ = 40, and $K_{Gal}$ = 0.055). Same color code as in (B).

Gal4p-mediated transcriptional activation. It was initially thought that activated Gal3p sequestered Gal80p in the cytoplasm, preventing it from its inhibitory role in the nucleus (Peng & Hopper, 2002). Later studies revised this view by showing that Gal3p molecules were not exclusively cytoplasmic (Jiang et al, 2009) and that forcing Gal3p to be mostly nuclear did not alter the kinetics of induction (Jiang et al, 2009). In addition, the slowness of the nucleocytoplasmic translocation of Gal80p, which depends both on transport rates and on the Gal4p:Gal80p dissociation rate, contrasts with the fast induction of transcription (Egriboz et al, 2011). This implies a direct role of Gal3p in promoting the dissociation of Gal80p from UAS. In addition, the galactokinase Gal1p (a paralog of Gal3p) can also act as a co-inducer of the regulatory circuit, presumably using similar mechanisms as Gal3p (Venturelli et al, 2012).

Our model covers the mRNA and protein species of three major players of GAL network induction: GAL1, GAL3, and GAL80, as well as of the reporter gene. We considered that promoters of each GAL gene could switch between an ON state (full transcription) and an OFF state (leaky transcription) at rates that depended on the concentration of Gal80 dimers, activated Gal3p dimers, and activated Gal1p dimers. The model is provided (computer Code EV1), and a detailed description of it is given in Materials and Methods and in Appendix Text S1. Most of the parameters of the model were fixed at values obtained from previous studies (Appendix Tables S1 and S2).

### Stochastic simulations reproduce the two types of induction observed experimentally

We first explored if our model captured the two types of responses of allele-replacement strains (i.e., binary and gradual). We ran stochastic simulations (Gillespie, 1977) that accounted for intrinsic and extrinsic sources of noise (see Appendix Text S1). We observed that tuning the parameters related to GAL3, while keeping all other parameters constant, was sufficient to modify inducibility and to obtain either a gradual (Fig 2B) or a binary (Fig 2C) response of the network at a given concentration of galactose. In the gradual system, the simulated single-cell trajectories were all similar; in the binary system, the simulated single-cell trajectories bifurcated, with a subset of cells having a stochastic lagging time before responding. The single-cell value of this lag time is directly correlated with the number of potential inducer proteins (Gal1p and Galp3p) present in the cell just before induction (Appendix Fig S2). This is in very good agreement with recent single-cell experiments on galactose induction (Stockwell & Rifkin, 2017). Note that a binary response is not necessarily a signature of steady-state bistability (Hermsen et al, 2011) but may represent a transient regime converging to a monostable ON state at equilibrium (see Appendix Text S1).

We then studied the response predicted by the model when stimulating the network with various concentrations of galactose while keeping model parameters constant (Appendix Fig S3). Inducibility increased with the concentration of galactose, with low concentrations causing a binary induction and high concentrations causing a gradual one.

### Two parameters related to GAL3 control network behavior

A detailed analysis of the model showed that inducibility of the system was mainly controlled by the average values of promoter switching rates $k_{on}$ and $k_{off}$ at the time of induction (see Materials and Methods, Appendix Text S1, and Figs S2 and S4). Rates $k_{off}$ depend only on GAL80 and are therefore invariant to GAL3 allelic variation. Rates $k_{on}$ depend on GAL3 in two ways: via Gal3p*, the amount of galactose-activated Gal3p, and via $K_3$, which corresponds to an effective concentration encompassing the dissociation constants of the Gal3p-Gal80p interaction and of Gal3p dimerization (see Appendix Text S1). Gal3p* is determined by the level of Gal3p and by parameter $K_{gal}$, which represents the typical concentration of galactose needed to efficiently activate Gal3p. While $K_{gal}$ was identifiable, several other GAL3-related parameters, such as those controlling the level of Gal3p, were not and we grouped them into a meta-parameter, $\rho_{Gal3}$, which we termed the *strength* of GAL3. $\rho_{Gal3}$ corresponds to the invert ratio between $K_3$ and the mean concentration of Gal3p at the time of induction, which depends on the leaky transcription rate, the translation rate and the degradation rates of GAL3 mRNA and protein product.

This formalism made the network sensitive to only two identifiable GAL3-related parameters, $K_{gal}$ and $\rho_{Gal3}$. At a fixed concentration of galactose induction, high $\rho_{Gal3}$ values correspond to high numbers of Gal3p dimers that can rapidly be activated to release Gal80 repression. The model predicted that high values of $\rho_{Gal3}$ would generate a gradual response (Appendix Fig S5A) because the number of potential activators was high enough in each cell to rapidly trigger the GAL1/GAL3-mediated positive feedback loop. In contrast, low values of $\rho_{Gal3}$ would generate a binary response (Appendix Fig S5B) because the number of activators is more stochastic, with many cells having too few initial Gal1p or Gal3p dimers to directly trigger the response. These cells need a lag time before fast activation (Fig 2B and C, and Appendix Fig S2). The other important parameter, $K_{gal}$, corresponds to a threshold of galactose concentration below which induction was limited and favoured a binary response, and above which induction was efficient and favoured a gradual response (Appendix Fig 3C). In summary, both $\rho_{Gal3}$ and $K_{gal}$ values can determine whether the network adopts a gradual or a binary response at a given concentration of galactose induction.

### Linking GAL3 alleles to specific parameter values

We first examined if our model could capture a known functional alteration of the GAL3 gene. The ATP and galactose binding pocket of Gal3p was previously described (Lavy et al, 2012). It contains an aromatic cage that encircles the adenine nucleotide of ATP. A mutation targeting this cage should therefore reduce the affinity of Gal3p for ATP. Galactose is localized more internally than ATP, suggesting that the sequential activation of Gal3p starts first with galactose binding and then ATP binding. We analyzed how parameters of our model should be affected by a mutation reducing Gal3p:ATP affinity, and we found that $K_{gal}$ should increase, that $\rho_{Gal3}$ should decrease, and that the relative change to wild-type values should be more pronounced for $K_{gal}$ (Appendix Text S1). In contrast, if ATP binds prior to galactose, then $K_{gal}$ should also increase but $\rho_{Gal3}$ should remain unaffected. Using Crispr/Cas9, we replaced in the BY reference strain the W117 residue of the cage by either an alanine or a threonine, and we monitored the dynamics of transcriptional activation of the corresponding mutants (Appendix Fig S6A–C). The non-conservative W117T mutation

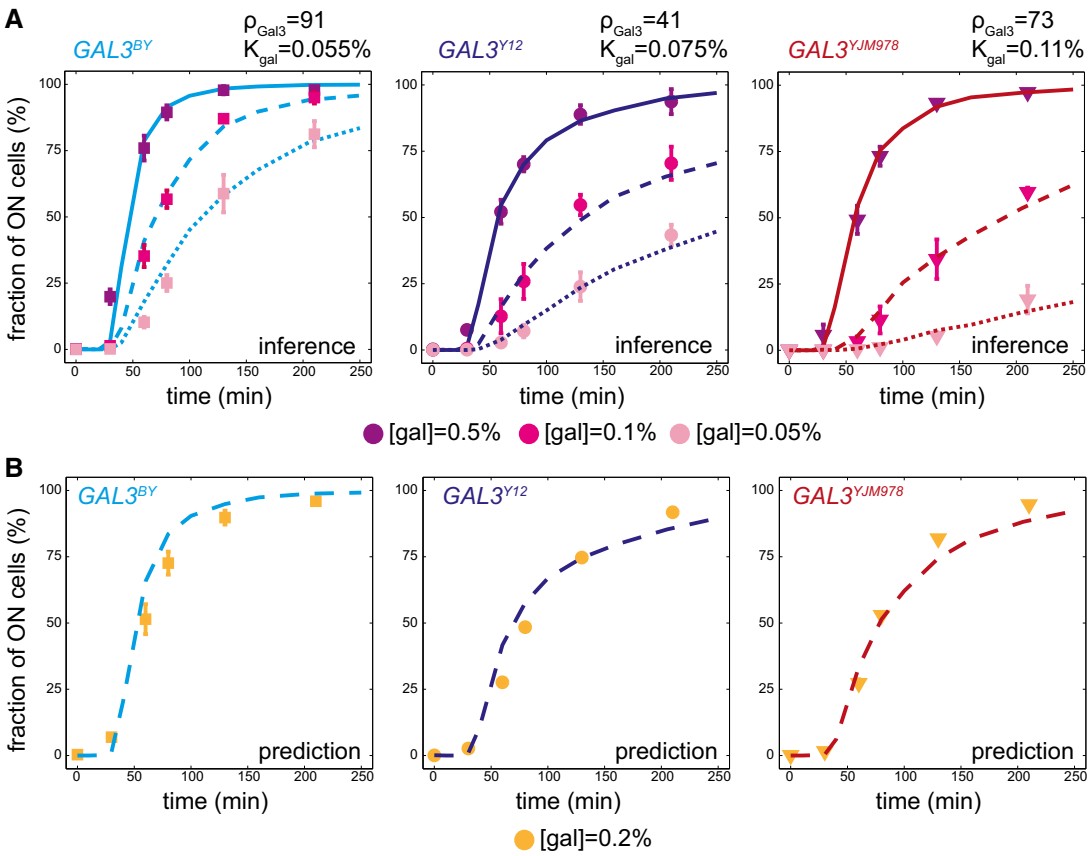

**Figure 3. Strain-specific training of the model and validation.**

A  Model fitting. Each panel corresponds to one strain carrying the indicated *GAL3* allele. Inducibility was measured by flow cytometry (data points $\pm$ s.e.m., $n = 6$) after stimulating cells with three different concentrations of galactose (points colored according to the concentration). For each strain, these data were used to fit the GAL3-dependent parameters $\rho_{Gal3}$ and $K_{Gal}$. Inferred parameter values are shown. Lines in plain (resp. dashed and dotted) represent the inducibility predicted by the model at [gal] = 0.5% (resp. 0.1 and 0.05%).

B  With the parameters inferred in (A), we use the model to predict the inducibility of each strain at a galactose concentration of 0.2% (lines), and this prediction was compared to experimental measures (dots $\pm$ s.e.m., $n = 2$).

fully abolished the response, even at very high concentrations of galactose. The W117A mutation profoundly reduced activation and caused a binary response. We then inferred parameters $\rho_{Gal3}$ and $K_{gal}$ for the wild-type strain and for the W117A mutant. This was done by selecting a set of parameters that minimized a global chi-squared score of deviation between the measured and predicted fractions of induced cells at different times after induction and for the different galactose concentrations (for details, see Materials and Methods and Appendix Text S1). Inferred $K_{gal}$ value was about 10 times higher for the mutant than for the wild type, and $\rho_{Gal3}$ was reduced by about threefold (Appendix Fig S6D). This fully agreed with the expected sequence of activation of Gal3p (galactose first, ATP second).

We then used our model to study natural *GAL3* alleles. We measured the transcriptional response of the *GAL3^BY*, *GAL3^Y12*, and *GAL3^YJM978* strains at different galactose concentrations (0.05, 0.1, and 0.5%). This confirmed that inducibility increased with galactose concentration (Fig 3A). We used these experimental data as above to infer parameters $\rho_{Gal3}$ and $K_{gal}$ for each of the three strains. To evaluate the usefulness of the inferred parameter values, we used

the fitted model to predict the behavior of each strain at a galactose concentration that was not used for model training (0.2%) (Fig 3B). Finally, to test model predictions, we experimentally monitored *GAL3^BY*, *GAL3^Y12*, and *GAL3^YJM978* induction at 0.2% galactose. Without any additional fitting procedure, we observed that inducibility (fraction of activated cells over time) differed between strains in a way that was entirely consistent with model predictions. Thus, the differences among parameter values assigned to the different natural *GAL3* alleles are relevant outside the specific experimental conditions used for parameters estimation.

## Natural *GAL3* alleles map to distinct locations of the parameter space

We sought to classify *GAL3* alleles based on the parameter values assigned to them. We made experimental measurements on two additional strains (*GAL3^NCYC361* and *GAL3^DBVPG1788*), and we determined best-fit $\rho_{Gal3}$ and $K_{gal}$ values to them as for the three strains described above. Appendix Fig S7 shows these data and the corresponding fitted models. Figure 4A and B shows the obtained

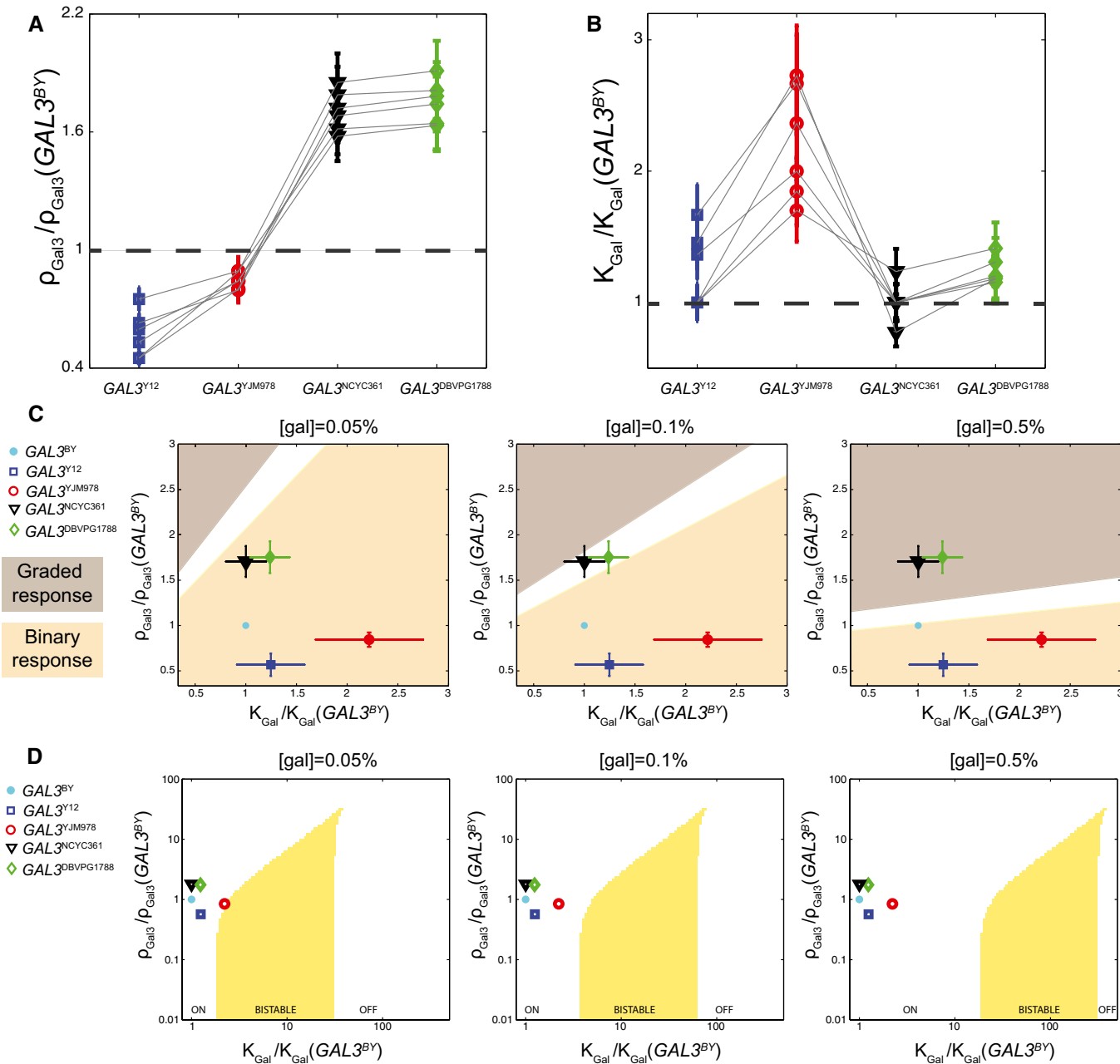

**Figure 4.  *GAL3* alleles map to distinct locations of the parameter space.**

A, B   Parameter values obtained by fitting the model to experimental data collected on five strains at three concentrations of the inducer ([gal] = 0.05, 0.1, and 0.5%). Six independent fits were performed (one per gray line). For each one, different values of GAL3-independent parameters were chosen (see Appendix Text S1), and parameters $\rho_{Gal3}$ (A) and $K_{gal}$ (B) were estimated for each strain. Dots represent their value for the indicated strain, relative to the value estimated for the $GAL3^{BY}$ strain. Error bars: uncertainty on parameter estimation for each inference (see Materials and Methods).

C   Phenotypic landscape predicted by the model. At defined concentrations of the inducer ([gal]), the values of $\rho_{Gal3}$ and $K_{gal}$ determine whether the response is gradual (brown) or binary (pink). The white zone is an intermediate region where the distinction between gradual and binary is unclear. Using parameters inferred in (A) and (B), alleles are mapped to the landscape (colored dots). Error bars: standard deviation of the six distinct estimations.

D   Bifurcation diagram of the deterministic description of the network at steady state. Yellow: region of the ($\rho_{Gal3}$, $K_{gal}$) space where the system is bistable at the indicated galactose concentration. Parameters: $\rho_{Gal1} = 100$, $\rho_{Gal80} = 250$, others as in Appendix Table S1. Symbols: positions of the five natural GAL3 alleles as in (C).

parameters, $\rho_{Gal3}$ and $K_{Gal}$, normalized by the corresponding values of our reference strain $GAL3^{BY}$. Different data points represent results obtained by applying the inference process to models with different GAL3-independent parameters (see Appendix Text S1).

The fold change of a parameter between two different strains is indicative of the functional nature of the genetic variations between the two *GAL3* alleles. In agreement with model predictions (Appendix Fig S3), we observed that more gradual strains

($GAL3^{NCYC361}$ and $GAL3^{DBVPG1788}$) display a high GAL3 strength $\rho_{Gal3}$ and a low "typical" galactose concentration $K_{Gal}$. Interestingly, we observed that $\rho_{Gal3}$ and $K_{Gal}$ can be decorrelated. In particular, although both $GAL3^{YJM978}$ and $GAL3^{Y12}$ strains were binary responders at all galactose concentrations tested, the model attributed this behavior to different functional effects: a low sensitivity to galactose (high $K_{gal}$) for the Gal3 protein originating from YJM978 and a reduced strength of the $GAL3$ gene originating from Y12. Thus, the induction specificities of the strains can be attributed to distinct GAL3-related parameters.

To address the direct relationship between the network properties (gradual or binary response) and the GAL3-related parameters, we positioned each of the tested strains within a phenotypic landscape according to their relative $\rho_{Gal3}$ and $K_{Gal}$ parameters (Fig 4C). According to our model, $\rho_{Gal3}$ and $K_{Gal}$ parameters are sufficient to predict the behavior (gradual or binary) associated with a given $GAL3$ allele at a given concentration of galactose. As an illustration of these predictions, we specifically observed the dynamics of transcriptional activation of the network for the strain $GAL3^{DBVPG1788}$ (Appendix Fig S8). The position of the $GAL3^{DBVPG1788}$ allele on the phenotypic landscape corresponded to a transient binary activation at low concentration ([gal] = 0.05%) converted into a gradual response at higher concentration ([gal] = 0.1% and [gal] = 0.5%). Importantly, mathematical analysis of the system at steady state indicated that all strains should reach a monostable ON state at equilibrium (Fig 4D and Appendix Text S1). Thus, although a binary regime of induction is observed in some strains, this is a transient regime that should eventually convergence to monostability after a very long time (> 10 h).

## Variation in induction dynamics is consistent with variation in diauxic shift decision

The physiological relevance of the GAL network regulation is to switch from the consumption of glucose (the preferred carbon source) to the consumption of galactose when glucose supply is running out. This diauxic switch is controlled not only by galactose induction but also by glucose-mediated repression. When both sugars are present, their relative concentration ratio determines whether cells activate the switch or not (New *et al*, 2014; Escalante-Chong *et al*, 2015; Venturelli *et al*, 2015). At some ratio values, only a fraction of the cells are induced, even at steady state. Given this dual regulation, the propensity of a strain to activate GAL metabolism can be quantified by measuring the fraction of induced cells after a prolonged period (8 h) of simultaneous induction (by galactose) and repression (by glucose). If this measurement is repeated at a given concentration of galactose and various concentrations of glucose, a useful score can be computed (called "decision threshold" hereafter): the concentration of glucose needed to maintain half the population of cells in the repressed (OFF) state (Fig 5A). A high decision threshold corresponds to an early activation of GAL genes during the diauxic shift.

A previous study identified *GAL3* as an important genetic determinant for this decision: The concentration ratio at which cells turn GAL expression ON differs between strains harboring different natural alleles of *GAL3* (Lee *et al*, 2017). We asked if this variation, visible in the steady-state exposure to both glucose and galactose, was correlated with the variation observed on the dynamics of network

induction upon exposure to galactose only. We chose four strains that showed different decision thresholds because of different *GAL3* alleles (Lee *et al*, 2017; Fig 5B), and we monitored their dynamics of induction at three different concentrations of galactose (with no glucose). We then used our model to assign $\rho_{Gal3}$ and $K_{Gal}$ parameter values to each strain. Experimental data and model fitting are shown in Fig 5B and Appendix Fig S9. We used the inferred parameter values to visualize the four strains in the parameter space where binary and gradual responses upon stimulation at [gal] = 0.25% are delimited (Fig 5D). Remarkably, the properties of induction dynamics in absence of glucose were fully consistent with the decision threshold during diauxic shift from glucose to galactose. Strains having a low decision threshold, such as $GAL3^{YJM421}$, displayed a transient binary response, and strain $GAL3^{BC187}$ had a high decision threshold and responded gradually. Coordinates of strains in the parameter space indicate that $\rho_{Gal3}$ values are highly informative on the decision threshold (Fig 5D). Thus, mapping allelic variation to dynamic parameters of induction is also useful to understand trade-offs that are observed at steady state.

## A quantitative parameter change predicts a role of H352D SNP on Gal3:Gal80 complex formation

We noticed that, at position 352 of the Gal3p protein, all natural strains harbored an aspartic acid, whereas the reference laboratory strain BY harbored a histidine. This aspartic acid was also conserved in *Saccharomyces mikatae*, *Saccharomyces paradoxus*, and *Saccharomyces uvarum* protein sequences (Cherry *et al*, 2012). Given the prevalence of this aspartic acid, we hypothesized that a single H352D amino acid change could have consequences on Gal3p regulatory function.

To test this, we generated an artificial $GAL3^{BY-H352D}$ allele by introducing the H352D mutation in the $GAL3^{BY}$ strain and we monitored the dynamics of induction of the resulting strain. At similar concentrations of galactose, induction was faster for the modified strain than for the original strain (compare Fig 6A with 2A). We then used our model to make functional predictions. We fitted our model to experimental data of induction as described above for natural alleles. Induction dynamics of the modified strain were fully explained by preserving parameter $K_{Gal}$ and increasing $\rho_{Gal3}$ (Fig 6B). This suggested that the H352D mutation did not affect activation of Gal3p by galactose but rather the strength of Gal3p, which summarizes six biochemical features: the basal level of *GAL3* transcription prior to induction, its translation and degradation rate, the degradation rate of its coding mRNA, its capacity to homodimerize, and the affinity of activated Gal3p for Gal80p.

How the implicated SNP could change either the leaky transcription level prior to induction or the transcription rate during induction is difficult to imagine. In addition, the amino acid change was not surrounded by any particular peptide motif, nor was it located at the extremity of the protein. This did not support for an effect on translation or degradation rates. Thus, the most plausible interpretation of the parameter change of the model was that the H352D modification would increase either the capacity of Gal3p* to dimerize or the affinity of the Gal3p* dimer for Gal80p.

To explore these possibilities, we analyzed the structure of the heterotetramer [Gal3p*]$_2$-[Gal80p]$_2$ that was previously solved (Lavy *et al*, 2012). We made three important observations. First,

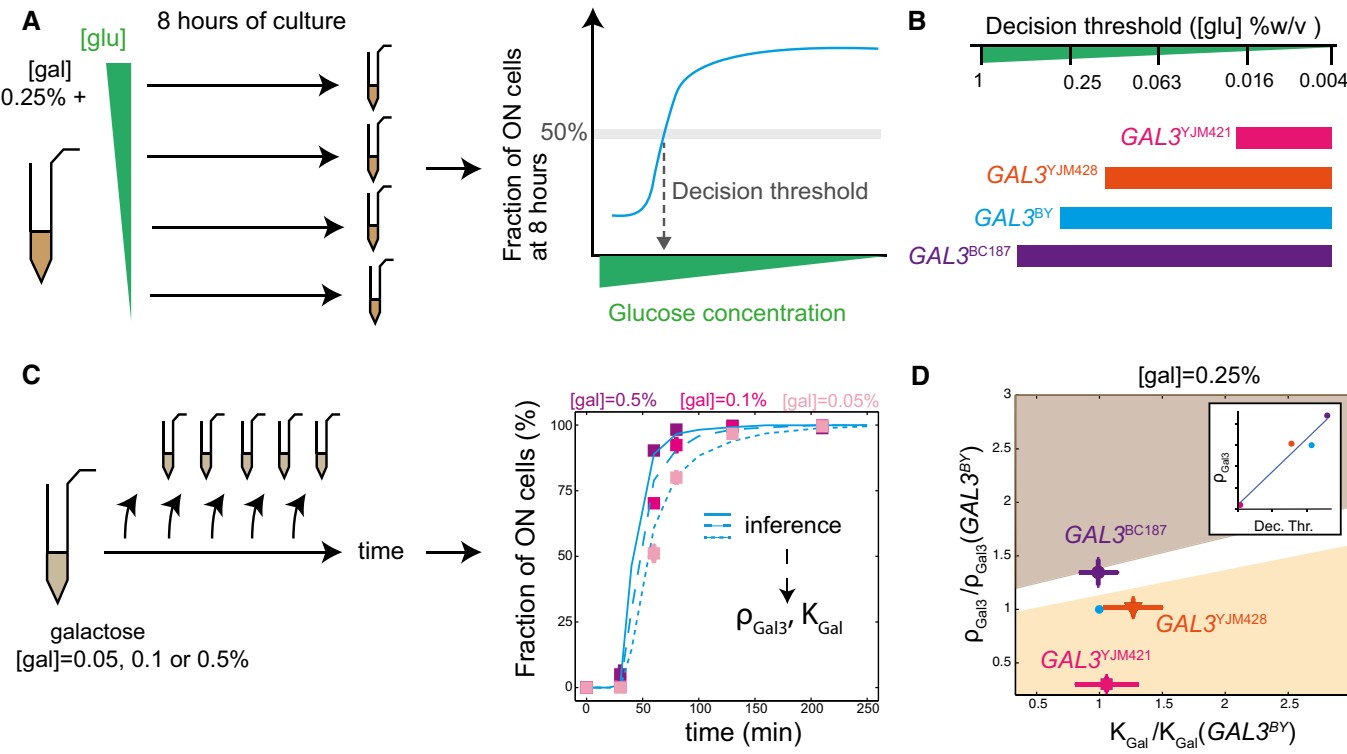

**Figure 5.  Relationship between inducibility and diauxic shift decision threshold.**

A   Schematic representation of decision threshold measurement. The decision threshold corresponds to the concentration of glucose at which 50% of the cells are induced in the presence of 0.25% galactose. The blue curve is theoretical and shown to explain how the fraction of ON cells depends on glucose concentration.

B   Decision thresholds for strains $GAL3^{BY}$, $GAL3^{YJM421}$, $GAL3^{YJM428}$, and $GAL3^{BC187}$ at [gal] = 0.25%.

C   Schematic representation of *GAL3* induction parameters determination.

D   Location of the *GAL3* replacement strains in the phenotypic landscape of the model at [gal] = 0.25%. Error bars: as in Fig 4C. Inset: $\rho_{Gal3}$ values as a function of the decision threshold, with dots corresponding to strains.

His352 is located at the binding interface of the Gal3p* dimer with the Gal80p dimer (Fig 6C), and distant from the pocket containing galactose and ATP. Secondly, it is spatially close to the Gal80p site where the acidic domain of Gal4p is known to bind (Thoden *et al*, 2008). Finally, the Gal80p dimer exhibits a positive electrostatic surface potential in the vicinity of Gal3p-His352, suggesting that the replacement of the neutral His352 by a negatively charged aspartic acid would stabilize the Gal3p*-Gal80p complex. Stabilization refers here to a gain in thermodynamic stability relative to the Gal4p-Gal80p complex, or in other words, to a decrease in Gibbs free energy change ($\Delta G_{sub}$) for the substitution of the Gal4p dimer by the Gal3p dimer as binding partner of the Gal80p dimer. A molecular dynamics simulation of the Asp352 mutant (in a model system of the Gal3p*-Gal80 complex) indicates that two positively charged amino acids, Gal3p-Arg362 and Gal80p-Lys287, are able to form direct salt bridges with Asp352 (Fig 6D). These attractive interactions of Asp352 with its environment are, however, expected to be partially canceled out by repulsive interactions with the less proximate, negatively charged amino acids Gal3p-Glu363 and Gal80p-Glu348 (Fig 6D). Also, the polar solution (water + counter ions) could partially reduce the stabilization effect of the H352D mutation because residue 352 is better solvated in the Gal3p* dimer than in the Gal3p*-Gal80p tetramer. Thus, to quantify a possible stabilization effect of the H352D mutation, we computed the change in the

Gibbs free energy difference, $\Delta\Delta G_{sub} = \Delta G_{sub}^{D352} - \Delta G_{sub}^{H352}$, with the aid of the thermodynamic cycle depicted in Fig 6E. The actual free energy calculations (see Materials and Methods) yielded $\Delta\Delta G_{sub} = -2.8 \pm 0.9$ kcal/mol, which indicates that the H352D mutation indeed increases the thermodynamic stability of the Gal3p*-Gal80p complex with respect to the Gal4p-Gal80p complex. Thus, as predicted by the dynamic model of network induction, the H352D mutation increases the cellular response by facilitating the formation of the complex.

## Discussion

We experimentally monitored the induction dynamics of the yeast GAL network in the context of natural genetic variation at the *GAL3* gene. This revealed that *GAL3* natural variation is sufficient to convert a gradual induction into a binary one. We built a stochastic model of the network and used it to link *GAL3* alleles to functional network parameters. This approach discriminated alleles that increased the strength of activated Gal3p (e.g., of strains NCYC361 and DBVPG1788) from alleles that desensitized Gal3p to galactose activation (e.g., of strain YJM978). Alleles showing different glucose/galactose trade-offs at equilibrium displayed different dynamics of induction, and they were associated with different

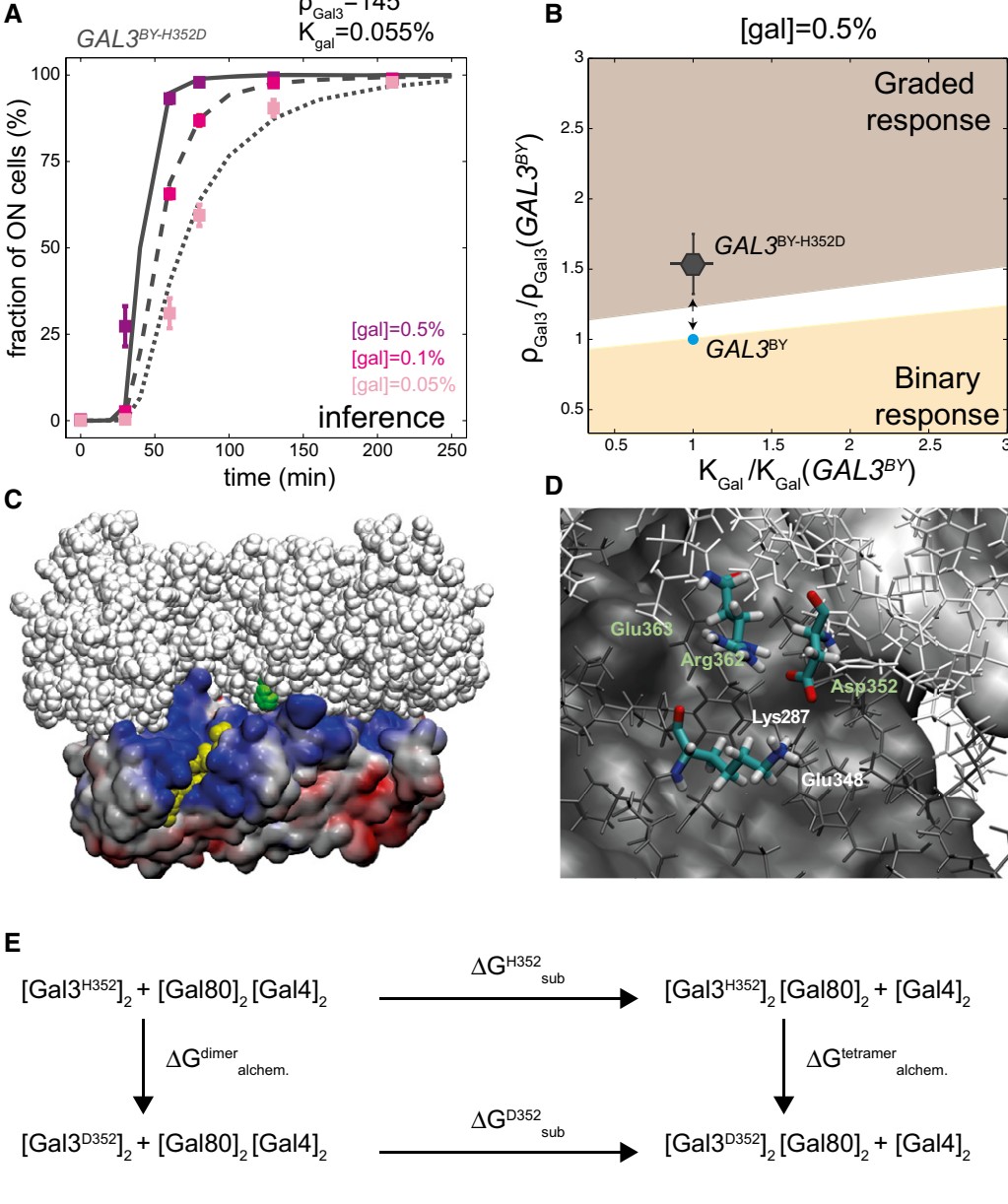

**Figure 6. Functional inference of the H352D variant of *GAL3*.**

A  Experimental acquisitions (dots ± s.e.m., *n* = 4) and model fitting (curves) of the induction dynamics of the *GAL3*^BY-H352D^ strain.

B  *GAL3*^BY^ (blue dot) and *GAL3*^BY-H352D^ (gray dot with standard deviation bars) strains localization in the phenotypic landscape of the model at [gal] = 0.5%. Arrows: phenotypic trajectory between the two alleles. Error bars: as in Fig 4C.

C  Structure of the tetrameric complex [Gal3p*]₂[Gal80p]₂ (PDB entry 3V2U). Residue His352 of one Gal3p unit is in the back side and not visible. The His352 residue of the other Gal3p unit is shown as green beads in the center; it is located at the binding interface of the Gal3p* dimer (white beads) and the Gal80p dimer (colored surface). Gal80p residues are colored according to their electrostatic surface potential from red (≤ −10 kT/e) to blue (≥ +10 kT/e). Yellow beads: The acidic activation domain of Gal4p was inserted in the complex by superimposition with crystal structure 3BTS. A similar insertion in the other Gal80p unit is in the back side and not visible. Created with VMD software.

D  Local stabilization of Gal3p-Asp352 by residues Gal3p-Arg362 and Gal80p-Lys287 in the [Gal3]₂[Gal80]₂ complex. Green and white labels refer to residues from Gal3p and Gal80p units, respectively. The figure shows a snapshot from a molecular dynamics simulation of the mutation H352D carried out for a model system of the complex (see Appendix Text S2). Atoms within 15 Å of residue 352 are shown as thin sticks in white (Gal3p) or dark gray (Gal80p). Remaining atoms are shown as a solid surface. Created with VMD software.

E  Thermodynamic cycle quantifying the energetic impact of the H352D mutation on the substitution of [Gal4p]₂ by [Gal3p]₂ as binding partner of [Gal80p]₂ ($\Delta G_{sub}$, horizontal arrows). This impact is measured as $\Delta\Delta G = \Delta G_{sub}^{D352} - \Delta G_{sub}^{H352}$, which equals to $\Delta G_{alchem}^{tetramer} - \Delta G_{alchem}^{dimer}$ (vertical arrows) because free enthalpy is a state function. These latter quantities correspond to the free enthalpy change for the alchemical (double) mutation of His > Asp in the Gal3p-Gal80p tetramer and in the Gal3p dimer, respectively, which were computed by alchemical free energy calculations (see Appendix Text S2).

strength of activated Gal3p. Our approach also predicted a functional effect of a single non-synonymous SNP that was validated by atomistic simulations of the binding interface between Gal3p and Gal80p dimers. These results provide further details on the yeast GAL system and, perhaps more importantly, they constitute a proof-of-concept of the feasibility and usefulness of linking genetic variants to model parameters.

## Genetic variability of the yeast GAL network

Our *in vivo* and *in silico* analyses of the induction kinetics of yeast GAL activation reveal properties of this system and how it is sensitive to genetic variation. The fact that an allelic exchange in *GAL3* was sufficient to convert a gradual response into a transient binary one is important because it shows that, although binary induction requires a polygenic architecture of transcriptional regulation, whether it takes place or not can be dictated by the genotype of a single gene.

Previously, several computational models of the network have been proposed, usually in an effort to understand the properties of the system at steady state (Acar *et al*, 2005; Apostu & Mackey, 2012; Venturelli *et al*, 2012). Particularly, they highlighted the important role of Gal3p-, Gal1p-, and Gal80p-mediated feedback loops. Our *in silico* analysis suggests that the gradual or binary kinetic response is mainly controlled by the initial number of repressors (Gal80p) and inducers (Gal1p and Gal3p), the efficacy of galactose to activate the inducers, and the efficiency of the activated inducers to release the effect of repressors. In particular, a low mean number of inducers at the time of induction may lead to high cell-to-cell variability in their actual number. Cells with few inducers (as compared to repressors) display a lag time before responding, leading to a binary response pattern at the population level. This prediction from our model is fully consistent with recent experiments that tracked the induction of the network at the single-cell level and showed that the initial concentrations of Gal1p and Gal3p are predictive of the transient bimodal response (Stockwell & Rifkin, 2017). We also observed that feedback loops were important to control the strength of cell-to-cell variability before induction (Gal80-mediated negative feedback) and the duration of lag times (Gal3/Gal1-mediated positive feedbacks), which agrees with the previous observation that disabling the Gal80p and Gal3p feedback loops can transform a gradual response into a binary one (Ramsey *et al*, 2006). Our results on GAL3 genetic variants also complement previous genetic manipulations of the feedback loops, where their effect on bimodality was tested by modulating promoter activities (Acar *et al*, 2005; Ramsey *et al*, 2006; Venturelli *et al*, 2012; Peng *et al*, 2015). Here, we showed that a non-synonymous variant affecting Gal3p:Gal80p interaction directly affects the dynamics of transient bimodality. This is a novel experimentally based observation that is totally coherent with the conclusions of Venturelli *et al* (2012) who showed computationally that steady-state bimodality of the network could rely on protein–protein binding affinities.

We also observed that genetic variations at *GAL3* could affect its propensity to be activated by galactose/ATP binding. In particular, the *GAL3*[YJM978] allele was associated with increased values of the $K_{Gal}$ parameter (more galactose needed for its activation). This allele harbored three non-synonymous SNPs: M179I, R312I, and H352D. As shown above, H352D is found in all natural alleles that we tested

and it therefore does not explain a change in $K_{Gal}$ specifically for *GAL3*[YJM978]. According to the structure of the Gal3p:Gal80p tetramer complexed with galactose and ATP (Lavy *et al*, 2012), the other two polymorphic sites do not map close to the pocket containing the ligands. Met179 is located at the surface of the complex, distant from any binding interface and distant from the bound galactose (30 Å) and ATP (25 Å). The mutational effect of the rather conservative amino acid change (methionine to isoleucine) on the $K_{Gal}$ parameter might therefore be negligible. In contrast, the non-conservative arginine to isoleucine mutation at site 312 could influence $K_{Gal}$ in several ways: First, the positively charged arginine contributes favorably to the binding of the negatively charged ATP through long-range electrostatic interactions. The charge-neutral Ile312 variant lacks this favorable interaction and may have lower affinity for ATP, thereby penalizing activation by the two ligands. Second, residues Arg312 of the two Gal3p units are in direct contact with each other, and the non-conservative R312I change may affect the dimerization of Gal3p. Lavy *et al* (2012) reported that, in absence of galactose, Gal3p is monomeric in solution and adopts an open conformation that differs from the conformation generating the Gal3p:Gal3p dimeric interface found upon interaction with Gal80p. If the R312I modification alters Gal3p dimerization, this could modify the overall activation by galactose because these processes are coupled.

We observed that genetic variation of the strength of activated Gal3p ($\rho_{Gal3}$), estimated from the dynamic properties of network activation, was correlated with variation of the glucose/galactose trade-off at steady state. This implies that the two traits co-evolve in natural populations of *S. cerevisiae*. Given the relatively short time scale of network induction, mild differences in the dynamics of activation alone are unlikely to cause fitness differences unless environmental galactose concentrations are highly dynamic. In contrast, variation in the sensitivity of the network to the ratio of external sugars corresponds to the triggering of an adaptive metabolic process, which is highly related to fitness even for slow environmental changes. The induction dynamics that we observed on short time scales are probably not themselves under selection, but they provide valuable information on the molecular mechanism affecting a fitness-related trait operating on longer time scales.

The H352D variant is interesting in this regard. At this position in Gal3p, a histidine residue was found in all laboratory strains (BY4741, CEN.PK, D273-10B, FL100, FY1679, JK9-3d, SEY6210, W303, X2180-1A, YPH499), while nearly all natural isolates as well as distant species possess an aspartic acid. Our results showed the importance of this aspartic acid for interaction with Gal80p, which suggests that its conservation in wild population results from purifying selection. The presence of slightly deleterious mutations in laboratory strains is well known. Examples from the reference strain BY/S288c include mutations in *AMN1* (Yvert *et al*, 2003), *BUL2* (Kwan *et al*, 2011), *ERC1* (Fehrmann *et al*, 2013), *FLO8* (Kron, 1997), *GPA1* (Yvert *et al*, 2003), and *HAP1* (Gaisne *et al*, 1999). These mutations likely resulted from a release of purifying selection caused by strong population bottlenecks when propagating yeast on petri dishes. As for the genes listed above, the implication for *GAL3* is that most mechanistic studies refer to a "wild-type" protein that is in fact a slightly hypomorphic allele not found in nature.

We also noted cases where the specificities of a *GAL3* allele in the context of the BY strain did not reflect the properties of the

donor strain. An extreme example of this was the $GAL3^{DBVPG1853}$ allele which improved the response of the BY strain (Fig 1D) while the DBVPG1853 strain itself did not respond at all to galactose (not shown), presumably because of genetic defects in other genes. Another example was the $GAL3^{Y12}$ allele which, when introduced in the BY strain, conferred a binary induction that was milder than the binary phenotype of the Y12 strain itself (Appendix Fig S10). This illustrates the importance of other loci than $GAL3$ that also contribute to the dynamics of network induction. Background-specific effects are common and should be taken into account when interpreting the functional impact of natural alleles in their original strain context (Gerke et al, 2010).

## Linking DNA variants to model parameters: feasibility and potential

We developed our approach using a model system, the yeast GAL network, which was an ideal context for investigation: Molecular players were well known, important network properties had been previously described, genetic engineering could be used to study the effect of a single gene in an otherwise isogenic background, and experimental measurements were relatively cheap. If network modeling had provided no added value in such a context, it would be hard to imagine how it could be useful in more complex frameworks. We report that it did: Observing different dynamics experimentally was not sufficient to make functional inferences, but combining data and modeling was. The concept is therefore fruitful and it is interesting now to consider how it can be extended to other biological systems.

First, it is important to realize that inference is based on the wealth of information contained in the dynamics of activation. Evidently, studying the system at equilibrium would not be sufficient. Mapping DNA variants to model parameters is therefore promising for systems where time-course data are available.

Second, even in the simple context of our study, not all parameters of the model were identifiable and it was necessary to aggregate several of them into a meta-parameter ($\rho_{Gal3}$). We admit that this constitutes a limit of the approach: When the H352D SNP was linked to this meta-parameter, additional assumptions were needed to infer biochemical effects. Similar difficulties will likely be encountered in other systems and the identifiability and sensitivity analysis of the model are therefore crucial to determine the nature of biological information that can be retrieved by the approach.

Third, our method here was to infer function and then to validate a prediction by exploring the structural data of a protein complex. Depending on the system under consideration and the data available, it may be judicious to reverse the approach: scanning protein structures first in order to identify variants modifying binding affinities and then studying these variants specifically using experimental measurements and model fitting. This way, a parameter change is first inferred from structural data and a dynamic model of the network then allows one to predict its phenotypic effect. The SAAP database (Al-Numair & Martin, 2013), which registers structurally relevant variants of human proteins, may constitute a very helpful resource to do this.

Fourth, while we based our approach on cell population distributions, tracking the response dynamics of individual cells over time is also possible (Stockwell & Rifkin, 2017) and can provide more information on the network response. In other contexts, such methods had been very useful to infer parameters associated with individual cells (Llamosi et al, 2016). A variant may then be associated with one parameter by a whole distribution of values, which likely carries more information than a single scalar value as presented here. In addition, such time-lapse acquisitions can provide high temporal resolution of gene induction (Aymoz et al, 2016).

Fifth, additional work is now needed to extend the approach to more than one gene. At the level of an entire network, the overall genotype of the individual is a combination of alleles. The number of such combinatorial genotypes of the network segregating in natural populations can be very large and mapping this diversity to the parameter space would be very interesting. In particular, models accounting for genetic changes might predict and explain genetic interactions (epistasis) within the network. The challenge to achieve this will likely reside in the number of free parameters: If the genotype is allowed to vary at too many genes, parameters cannot be constrained efficiently. Mapping variants one gene at a time, as we did here, and then in combination would maintain this necessary constraint while evaluating epistasis. A more difficult task would be to infer the contribution of genes that are external to the network while nonetheless affecting its behavior (e.g., by modifying widely transcription rates or the stability of proteins, or cross-talks with other networks). Studying these factors by our approach is only possible after they are identified and connected to the network. Their identification can be obtained by genetic mapping. For example, we recently identified a locus on yeast chromosome V that affects the variability of the GAL response at transient times of activation (Chuffart et al, 2016). Once identified, these factors must be integrated in the network model, which may be a complex task.

Network modeling is expected to help the development of personalized medicine and the fact that it is possible, in a yeast system, to personalize model parameters according to DNA variants is encouraging. Can the approach described here be applied to human variants? This requires overcoming several difficulties that could be avoided in our framework. First, most regulatory networks of human systems are incompletely known. Second, most of these networks comprise numerous genes, implying many model parameters and, possibly, too many degrees of freedom for adjustments and identifiability issues. The first task is therefore a careful identifiability and sensitivity analysis of the model and, as much as possible, a reduction of its complexity. The work of Zhao et al (2015) is encouraging in this regard. The authors studied the mitochondrial outer membrane permeabilization network controlling entry in apoptosis. Their model comprised ~50 parameters and ~20 molecular species, but the network critical behavior (bifurcation point) was sensitive to less than half of the parameters. The authors then searched for enrichment of cancer mutations in protein domains involved in molecular interactions and they used molecular dynamics simulations to estimate the affinity changes caused by these mutations. Interestingly, most mutations that were predicted to affect sensitive parameters of the model caused a significant change of affinity in the expected direction, illustrating that the model was able to highlight relevant vulnerabilities. Similarly, Nijhout et al (2015) studied a model of the folate-mediated

one carbon metabolism system. They reported that human mutations that strongly perturb enzymatic activities could have little phenotypic effect if they targeted parameters that are poorly sensitive. Another type of difficulties when studying human networks are experimental limitations: Manipulating human cells needs more time and funds than manipulating yeast; replacing alleles of specific genes is possible via CRISPR/Cas9 editing but the large physical size of human genes as well as the functional redundancy between paralogs can be problematic; and setting up dynamic experimental acquisitions is often not straightforward. Thus, applying our approach to a minimal network in human cells compatible with genetic editing and time-series acquisitions will probably constitute an important step in the near future.

# Materials and Methods

## Yeast strains and plasmids

The strains used in this study are listed in Appendix Table S3. Oligonucleotides are listed in Appendix Table S4. We used the strain BY4711 (GY145, isogenic to s288c) as BY reference strain. The $P_{GAL1}GFP$ reporter cassette was obtained from plasmid pGY338 previously described (Chuffart *et al*, 2016). pGY338 was linearized by NheI and integrated at the *HIS3* locus of BY4711 to create strains GY1648 and GY1649, two independent transformants. To replace endogenous $GAL3^{BY}$ allele by natural variants in GY1648 strain, we PCR-amplified the *TRP1-GAL3* locus of natural wild isolates using primers 1D28 and 1D56. The endogenous locus was then replaced by *in vivo* homologous recombination and positive transformants were selected on SD-TRP plates. $GAL3^{NCYC361}$, $GAL3^{K11}$, $GAL3^{Y12}$, $GAL3^{DBVPG1788}$, $GAL3^{DBVPG1853}$, $GAL3^{YJM978}$, $GAL3^{JAY291}$ were PCR-amplified from NCYC3451, NCYC3452, NCYC3445, NCYC3311, NCYC3313, NCYC3458 [wild isolates from the *Saccaromyces* Genome Resequencing Project, SGRP (Louis & Durbin, 2007; Liti *et al*, 2009)] and JAY291 (Argueso *et al*, 2009), respectively. The strains used to characterize the effect of natural variants on galactose response were GY1648, GY1689, GY1692, GY1695, GY1698, GY1704, GY1707, and GY1713, all isogenic to S288c except for $GAL3^{BY}$, $GAL3^{NCYC361}$, $GAL3^{K11}$, $GAL3^{Y12}$, $GAL3^{DBVPG1788}$, $GAL3^{DBVPG1853}$, $GAL3^{YJM978}$, $GAL3^{JAY291}$, respectively. Strains genotype was verified by PCR and either high-resolution melting curves, restriction fragment-length polymorphism typing, or sequencing. The *TRP1-GAL3* locus from BY strain was PCR-amplified with primers 1M95 and 1M96 and cloned into HpaI-linearized plasmid pALREP (Fehrmann *et al*, 2013) by homologous recombination in yeast, generating plasmid pGY409. The mutated $GAL3^{BY\text{-}H352D}$ allele was synthesized by GeneScript and subcloned into pGY409 using MscI-BstEII restriction sites, generating plasmid pGY418. The *TRP1-GAL3*$^{BY\text{-}H352D}$ locus was PCR-amplified from pGY418 using primers 1D28 and 1D56 and transformed into GY1649 to create strain GY2009. Genotype was validated by PCR and sequencing. Strains of Fig 5 were MPJ125-E06 ($GAL3^{BY}$), MPJ143-H01 ($GAL3^{YJM428}$), MPJ143-F01 ($GAL3^{YJM421}$), and MPJ125-A07 ($GAL3^{BC187}$), which were described in another study (Lee *et al*, 2017); they all derived from a S288c *hoΔ::GAL1pr-YFP-mTagBFP2-kanMX4; gal3Δ::hphNT1* parental strain. Strains GY2180 and GY2181 carrying mutations $GAL3^{W117A}$ and $GAL3^{W117T}$, respectively, were obtained using

CRISPR/Cas9: Oligonucleotides 1P29 and 1P30 were annealed and cloned in the SwaI, BclI sites of the pML104 plasmid (Laughery *et al*, 2015) to produce pGY514. Repair-templates were produced by PCR on pGY474 (a plasmid containing the wild-type GAL3 gene from strain BY) using primers 1P22 and 1J63 (for W117A) or 1P23 and 1J63 (for W117T). Strain GY1566 (Chuffart *et al*, 2016) was co-transformed with pGY514 and the repair template and plated on −URA selection medium. Transformants were verified by PCR with 1P21 and 1J63 followed by Sanger sequencing. They were cultured on YPD plates, and plasmid loss was verified by replica-plating on −URA plates.

## Galactose response measurements

Liquid cultures in synthetic medium with 2% raffinose (yeast nitrogen base w/o amino acids 6.7 g/l, raffinose 2%, dropout Mix 2 g/l, adjusted to pH = 5.8) were inoculated with a single colony and incubated overnight, then diluted to OD600 = 0.1 (synthetic medium, 2% raffinose) and grown for 3–6 h. The galactose induction experiments were carried out in 96-well sterile microplates using a Freedom EVO200 liquid handler (Tecan) equipped with a 96-channel pipetting head (MCA), a high precision 8-channel pipetting arm (LiHa), a robotic manipulator arm (RoMa), and a MOI-6 incubator (Tecan). All robotic steps were programmed in Evoware v2.5.4.0 (Tecan). Cells were resuspended in synthetic medium with 2% raffinose and the appropriate galactose concentration (0.01, 0.1, 0.2, and 0.5%) and grown for the desired time (from 0 to 250 min). Cells were then washed with PBS1X, incubated for 8 min in 2% paraformaldehyde (PFA) at room temperature, followed by 12 min of incubation in PBS supplemented with glycine 0.1 M at room temperature and finally resuspended in PBS. They were then analyzed on a FACSCalibur (BD Biosciences) flow cytometer to record 10,000 cells per sample. Each set of data is representative of the results of two independent experiments (each comprising three technical replicates).

Flow cytometry data were analyzed using the *flowCore* package from Bioconductor (Hahne *et al*, 2009). Cells of homogeneous size were dynamically gated as follows: (i) removal of events with saturated signals (FSC, SSC, or FL1 = 1023 or = 0), (ii) correction by subtracting the mean(FL1) at $t = 0$ to each FL1 values, (iii) computation of a density kernel of FSC, SSC values to define a perimeter of peak density containing 60% of events, (iv) cell gating using this perimeter, (v) removal of samples containing less than 3,000 cells at the end of the procedure, and (vi) correction of the data according to an eventual experimental bias during cytometer acquisitions. For the twelve time points (0, 10, 20, 30, 40, 60, 80, 100, 130, 160, 205, and 250 min) experimental design, the time course for a given strain was acquired on different plates on the flow cytometer. In order to correct an eventual plate effect, we systematically included 24 replicates on each plate acquired on flow cytometer. We then tested the fixed effect of plates using an ANOVA. The FL1 values of each cell were subsequently corrected according to the plate offset of the ANOVA. For the six time points (0, 30, 60, 80, 130, and 210 min) experimental design, all the time points being acquired on the same experimental plate, we did not apply the normalization filter. The GFP expression values presented here in arbitrary units were the FL1 signal of the retained cells (normalized for the plate effect, if required).

## Analysis of flow cytometry distributions

All statistical analyses were done using R (version 3.2.4).

### Calculation of the response amplitude

The response amplitude A was defined as the mean of $P_{GAL1}GFP$ expression in activated cells. First, for each strain, at each time point, we determined by eye if the $P_{GAL1}GFP$ distribution was unimodal $\left(f_{(X_{ALL})} = N(\mu_{ALL}, \sigma_{ALL})\right)$ or bimodal $\left(f_{(X_{ALL})} = f_{(X_{OFF})} + f_{(X_{ON})}\right)$. If the distribution was unimodal, we calculated: $A = \mu_{ALL}$. Otherwise, bimodal distributions were considered as mixtures of two normal distributions, such as: $f_{(X_{ALL})} = \rho_{OFF}N(\mu_{OFF}, \sigma_{OFF}) + \rho_{ON}N(\mu_{ON}, \sigma_{ON})$, with $A = \mu_{ON}$. We used the function mixtools::normalmixEM() to calculate A for mixture distributions.

### Calculation of inducibility

Inducibility was defined as the proportion of ON cells in the population. The threshold $t$ between OFF and ON cells was calculated as follows: (i) a subset of OFF cells (all cells acquired at $t = 0$ min) and ON cells (activated cells belonging to unimodal distributions, acquired at the latest time point of the experiments) was defined for each experiments, (ii) the mean and standard deviation were extracted from each OFF and ON normal distributions using the function mixtools::normalmixEM(), (iii) these parameters were used to determine $t$ such as $\mathbb{P}(X_{ON} < t) = \mathbb{P}(X_{OFF} > t)$, with $X_{ON}$ the observed fluorescence FL1 in ON_cells and $X_{OFF}$ the observed fluorescence FL1 in OFF_cells, (iv) we finally calculated $I = \frac{nb\_cells(FL1 > t)}{nb\_cell(total)}$ for each time point, for each strain.

## Stochastic modeling

We model the stochastic gene expression of *GAL1, GAL3, GAL80*, and of the reporter gene (under a *GAL1* promoter). For each gene, we account for the status of the promoter (ON/OFF) and for the production and degradation of mRNAs and proteins. In addition, for the reporter gene, we account for the maturation of the fluorescent protein. The promoter switching rate from ON to OFF for gene $i$ is driven by GAL80p: $k_i^{off} = k_o^{off}\left[\left(\frac{Gal80p}{K_{80}}\right)^2\right]^{n_i}$ with $n_i$ the number of strong GAL4p binding sites in the promoter. We assume that GAL80p represses transcription via its dimerized form (with $K_{80}$ encompassing the dimer dissociation constant). The promoter switching rate from OFF to ON is driven by GAL3p and Gal1p: $k_i^{on} = k_o^{on}\left[\left(\frac{Gal1p^*}{K_1}\right)^2 + \left(\frac{Gal3p^*}{K_3}\right)^2\right]^{n_i}$ with $Galp^* = Galp\left(\frac{[gal]/K_{gal}}{1 + [gal]/K_{gal}}\right)$ the number of activated proteins at a given galactose concentration [gal] ($K_{gal}$ being the galactose dissociation constant). Here also, we assume that activated Gal3p and Gal1p are mainly found as dimers. $K_1$ and $K_3$ encompass the dimer dissociation constants as well as the affinity of activated proteins for Gal80p. For a detailed description of the model, see Appendix Text S1. Most of the parameters of the model (except $K_1$, $K_3$, $K_{80}$, and $K_{gal}$) were fixed based on the literature (see Appendix Table S1 in Appendix Text S1). The model had 7 GAL3-dependent parameters: $\alpha_3$ (leaky transcription rate), $\gamma_3$ (translation rate), $\beta_3$ (mRNA degradation rate), $\mu_3$ (protein degradation/dilution rate), $\Delta\alpha_3$ (full transcription rate), $K_3$, and $K_{gal}$. The phenotypic response of a strain (gradual vs. binary) at a given galactose concentration mainly depends on $K_{gal}$ and on the strength of GAL3 defined by $\rho_{Gal3} = \alpha_3\gamma_3/(\beta_3\mu_3K_3)$ (see main text, Appendix Text S1 and

Appendix Fig S12). For a given set of parameters, the stochastic dynamics of galactose induction was simulated using the stochastic simulation algorithm from Gillespie (Gillespie, 1977). The system was first allowed to reach steady state at [gal] = 0. At $t = 0$, galactose is introduced and the parallel—independent—evolution of 5,000 cells is monitored during 250 min of real time. The model is provided in computer Code EV1.

## Parameter inference

For a fixed set of GAL3-independent parameters, predictions for various values of GAL3-dependent parameters $\rho_{Gal3}$ and $K_{gal}$ were performed at three different galactose concentrations (0.05, 0.1, and 0.5%). Parameters were sampled from a 2D logarithmic-grid encompassing the region of interest. Then, for each strain, a global chi-squared score between the experimental data and the corresponding model predictions integrating the three concentrations were minimized to infer $\rho_{Gal3}$ and $K_{gal}$. Uncertainties on the parameters reflect the size of the sampling parameter grid. Parameter inference was repeated six times for different values of GAL3-independent parameters (see Appendix Text S1).

*Molecular dynamics simulations* for free energy calculations were carried out as described in Appendix Text S2 and Fig S11.

## Data availability

All flow cytometry raw data files can be downloaded from http://flowrepository.org under accession number FR-FCM-ZY6Y.

**Expanded View** for this article is available online.

## Acknowledgements

We thank Orsolya Symmons for discussions, Olivier Gandrillon for critical reading of the manuscript, Nicolas Rochette for sequence analysis, Sandrine Mouradian and SFR Biosciences Gerland-Lyon Sud (UMS344/US8) for access to flow cytometers and technical assistance, the Pôle Scientifique de Modélisation Numérique and the CIMENT infrastructure (supported by the Rhône-Alpes region, Grant CPER07_13 CIRA) for computing resources, BioSyL Federation and Ecofect LabEx (ANR-11-LABX-0048) for inspiring scientific events, developers of R, bioconductor and Ubuntu for their software, and two anonymous reviewers for their comments. This work was supported by the European Research Council under the European Union's Seventh Framework Programme FP7/2007-2013 Grant Agreement no 281359. DJ acknowledges program AGIR of University Grenoble-Alpes and the Institut Rhône-Alpin des Systèmes Complexes for funding. M. Spr. was supported by grants NSF 1349248 and RO1 GM120122-01 from the National Science Foundation (USA).

## Author contributions

Performed experiments: MR, HD-B, EF, ME, AB; Contributed analysis tools: MR, FC, DJ, and GY; Contributed reagents: MSpr; Developed and evaluated pilot versions of the model: FC and FP; Conceived, implemented and used the model: DJ; Performed molecular dynamics simulations: MSpi; Interpreted results: MR, MSpr, MSpi, DJ and GY; Conceived and designed the study: GY; Wrote the paper: MR, DJ and GY.

## Conflict of interest

The authors declare that they have no conflict of interest.

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
