## [Review Process File · Molecular Systems Biology]

Assigning function to natural allelic variation via dynamic modeling of gene network induction

Magali Richard, Florent Chuffart, H el ene Duplus-Bottin, Fanny Pouyet, Martin Spichty, Etienne Fulcrand, Marianne Entrevan, Audrey Bartheleix, Michael Springer, Daniel Jost and Ga el Yvert

Review timeline:

Submission date:	7 June 2017
Editorial Decision:	18 July 2017
Revision received:	22 November 2017
Editorial Decision:	7 December 2017
Revision received:	15 December 2017
Accepted:	18 December 2017

Editor: Maria Polychronidou

Transaction Report:

1st Editorial Decision

18 July 2017

Thank you again for submitting your work to Molecular Systems Biology. We have now heard back from the two referees who agreed to evaluate your study. As you will see below, the reviewers raise a series of concerns, which we would ask you to address in a revision of the manuscript.

The reviewers' recommendations are rather clear so I think that there is no need to repeat all the points listed below. A particularly important point was raised by reviewer #1 (major point #3) and refers to the need to better clarify/demonstrate how the model can be applied to assign function to allelic variants.

REVIEWER REPORTS

Reviewer #1:

Linking genotypes-to-phenotypes is a particularly complex problem in biology and even more so for dynamical systems. One promise of systems biology is that by modeling cellular systems using parameters that can be tuned by mutations, one would be able to predict the systems responses to mutations and thus phenotypes, including for instance diseases and drug responses. Very few model systems allow to develop such approaches because they are not fully described, let alone understood. The yeast GAL pathway is a canonical transcriptional pathway that has been studied for decades and that was shown recently to be polymorphic in the budding yeast *Saccharomyces cerevisiae*. Some strains show a graded response and others a transient binary response. Variation in

sensitivity to galactose among strains was shown to associate with variation in GAL3, a gene coding for a protein that binds galactose and ATP and that releases the repression on Gal4p that represses GAL genes. Richard and colleagues use this system to isolate variation in GAL3 and use it to examine whether modeling can be used to dissect how mutations in this gene affect the dynamics of the response. The authors demonstrate that this is possible and focus on one particular amino acid substitution on Gal3p that most likely affects its binding to Gal80p.

General remarks

This manuscript is a conceptual advance in our understanding of genotype-phenotype maps. Many studies have used the GAL system to improve our understanding of cell signaling and regulation and this report builds on this knowledge by examining how natural variation in the pathways affect its dynamics. Given all we know about genetic polymorphism in humans and other species, one can consider that to be useful, models of pathway dynamics have to be able to accommodate parameters that will be affected by variation in gene regulation and protein functions. This manuscript is therefore an important step forward and I would expect it to be of major interest for people in the field of systems biology and systems genetics, and this, beyond model systems.

Major points

- I cannot speak to the specifics of the mathematical aspects of the model. However, I believe that the experiments are sound and valid and appear to be in line with previous experiments that have examined the dynamics of induction of the GAL network. The results are well presented and made accessible. The demonstration for instance that genetic variation in GAL3 is sufficient to convert a gradual response into a binary one is very interesting and has a broad impact on this field, in which people usually assume that these two dynamics require very different underlying architecture. What is shown here is that few amino acid substitutions are sufficient to go from one type to the other. This finding could occupy more space in the paper.
- It would be useful to know what the pGal1-GFP expression actually reflects in the GAL pathway. Since what eventually matters is the response of the cell to galactose (growth?), one may want to know what is critical for this response, for instance is it the amount of a specific enzymes or set of enzymes? and how reading the pGal1-GFP activity reflects this response. Also, it would be useful to know what the effect of the destabilized GFP is on the inferences made. I understand that this is used to assess transcriptional output and not protein accumulation. However, if proteins of the pathway and long-lived, could we expect some of the features of the GFP response such as bi-modality to be irrelevant to the actual activity of the pathway itself? Could long-lived proteins attenuate this bi-modality and make it unimodal?
- One major weakness I see from the study is that the conclusion is on the use of dynamical modeling to assign function to natural allele variation but the example chosen affects the binding of Gal3p to Gal80p. The analyses and some statements in the abstract for instance (line 38) give the impression that this stands out from the modeling directly and that the model could lead to this. However, the impression from the paper is that this could have been found simply by looking at the protein complex structure and conservation of the residues, and that the dynamical modeling was not instrumental in identifying this mutation as critical for binding. The title also suggests that the function of the alleles that are altered can be discriminated from the model but the model uses two parameters Kgal and pgal3 that themselves can be tuned in many different ways by mutations and it is not even clear how these two parameters can be discriminated by the model. I guess this issue can be resolved by adjusting the scope of the paper so that it fits the actual finding. Another approach would be to create targeted GAL3 alleles using mutations of known or predicted effects to show that these parameters can be tuned (or not) independently and how one could for instance possibly dissect our all different parameters that go into the "strength" of Gal3p.

Minor points

- Line 280: Does using an intermediate galactose concentration actually reflects a condition that is not used for parameter estimation?
- References on the prediction of functional impacts of SNPs from line 66 to 69 appear to be from a few years ago (ref 2 for instance). It may be worth including more recent ones.
- Line 79 to 82. Since the work is put into the context of personalized medicine, i.e. using a patient's genotype and other information to predict, diagnose and treat diseases, I would be useful to briefly explain what we mean in practical terms by 'identifiable', sufficiently "constrained" and parameters that can be "reliably inferred". This is particularly important with respect to other approaches based

on machine learning for instance in which we can make predictions without knowing how the system works, which is thought to be sufficient for many medical applications.

- Line 198. Many statements are made about the network as if they were universally true, for instance that Gal80p forms an homodimer etc. I suspect that this knowledge derives from studies performed on a single strain. In the light of the diversity uncovered in this study, would it be useful to specify this somewhere? In some ways, because of polymorphism, there is not "a" GAL pathway, there are many different ones.

- Are parameter values used and reported in Table S1 derived from the same genetic background? If not, does it matter for this work?

- It would have been useful to know how the different GAL3 alleles affect the network dynamics when in their own genetic background, i.e. comparing the endogenous GAL3 allele in a background with that of the GAL3 BY alleles. It could have illustrated the contribution of the alleles versus other loci in differentiating the strains.

- Figure 2. Is Gal4p not on the figure?

Reviewer #2:

The authors study how alleles of the GAL3 gene affect the induction kinetics of the GAL network and fit binding parameters to the observations. They observed two different types of responses with the different alleles: binary and graded. The model describing the above responses in different conditions should be worked out more precisely before publication:

1. Binary / graded responses.

1A.(minor) The terms should be used more consistently. For, example the authors write:

"Inducibility increased with the concentration of galactose, with low concentrations causing a probabilistic induction (binary) and high concentrations a deterministic one (gradual)."

Even a graded induction can be stochastic. Therefore, the authors should use only phenomenological descriptions, i.e. binary and graded (or gradual), and omit terms that are used to describe model types (deterministic models vs probabilistic - stochastic models).

1B. (major). Binary response is usually - but not always - a sign of an underlying deterministic bistability. For example, a binary response can arise in the absence of bistability [PMID: 22125482]. The authors should determine if the deterministic model corresponding to the stochastic model with the fitted parameters displays monostability or bistability (ie. the model consisting of the ordinary differential equation has two stable solutions). Then it will be possible to show if the binary and graded responses correspond to bistable and monostable parts of the parameter space. If the binary response is associated with a monostable solution, this may be a sign of a slow transient [PMID: 27498164]. This may be the case since the authors indicate for some alleles that: "Strains having a low decision threshold, such as GAL3YJM421, displayed a transient binary response, and strain GAL3BC187 had a high decision threshold and responded gradually.". The transient response can be often slower than usually expected. In fact, steady-state condition is rarely attained in realistic stochastic systems.

2. Galactose / Glucose conditions: The expression of GAL4 depends on glucose (e.g. PMID: 915298, PMID: 28333434). However, the authors consider the GAL4 expression to be constant. Of course, the metaparameters can be fitted with constant GAL4 expression because the experiments were performed in galactose / raffinose. However, the glucose experiments (e.g. Fig. 5) require the inclusion of the GAL4 response to glucose to see if the Kgal and RhoGal3 parameters maintain their relationship with respect to the glucose threshold.

The authors may also show the GAL4 response by measuring GAL4 RNA.

3. Consistent distinction between the direct and indirect fitting and observation.

The authors use a precise definition of the metaparameter (e.g. "Second, even in the simple context of our study, not all parameters of the model were identifiable and it was necessary to aggregate several of them into a meta-parameter".) to indicate that it is an "indirect parameter". However, they cite observations as if they were direct even if they are indirect. E.g. " that the dynamics of nucleocytoplasmic trafficking were too slow to explain the fast induction of transcription²⁴."

However, the binding and shuttling in those experiments are lumped and are not distinguished. The Gal80-GFP translocation to the cytoplasm depends both on the transport rate and the Gal4-Gal80 dissociation rate. Two latter could have been eliminated in Gal4 deletion strains. Thus, the above

observation can be interpreted in terms of a metaparameter. Therefore, I suggest a more precise formulation in such cases.

1st Revision - authors' response

22 November 2017

We thank the reviewers for their helpful comments.

Reviewer #1:

*Linking genotypes-to-phenotypes is a particularly complex problem in biology and even more so for dynamical systems. One promise of systems biology is that by modeling cellular systems using parameters that can be tuned by mutations, one would be able to predict the systems responses to mutations and thus phenotypes, including for instance diseases and drug responses. Very few model systems allow to develop such approaches because they are not fully described, let alone understood. The yeast GAL pathway is a canonical transcriptional pathway that has been studied for decades and that was shown recently to be polymorphic in the budding yeast *Saccharomyces cerevisiae*. Some strains show a graded response and others a transient binary response. Variation in sensitivity to galactose among strains was shown to associate with variation in GAL3, a gene coding for a protein that binds galactose and ATP and that releases the repression on Gal4p that represses GAL genes. Richard and colleagues use this system to isolate variation in GAL3 and use it to examine whether modeling can be used to dissect how mutations in this gene affect the dynamics of the response. The authors demonstrate that this is possible and focus on one particular amino acid substitution on Gal3p that most likely affects its binding to Gal80p.*

General remarks

This manuscript is a conceptual advance in our understanding of genotype-phenotype maps. Many studies have used the GAL system to improve our understanding of cell signaling and regulation and this report builds on this knowledge by examining how natural variation in the pathways affect its dynamics. Given all we know about genetic polymorphism in humans and other species, one can consider that to be useful, models of pathway dynamics have to be able to accommodate parameters that will be affected by variation in gene regulation and protein functions. This manuscript is therefore an important step forward and I would expect it to be of major interest for people in the field of systems biology and systems genetics, and this, beyond model systems.

Major points

• I cannot speak to the specifics of the mathematical aspects of the model. However, I believe that the experiments are sound and valid and appear to be in line with previous experiments that have examined the dynamics of induction of the GAL network. The results are well presented and made accessible. The demonstration for instance that genetic variation in GAL3 is sufficient to convert a gradual response into a binary one is very interesting and has a broad impact on this field, in which people usually assume that these two dynamics require very different underlying architecture. What is shown here is that few amino acid substitutions are sufficient to go from one type to the other. This finding could occupy more space in the paper.

This finding now explicitly appears in the corresponding subsection header of the results (line 165). Its importance is also specifically mentioned in the revised discussion (lines 438 and 454-457).

• It would be useful to know what the pGal1-GFP expression actually reflects in the GAL pathway. Since what eventually matters is the response of the cell to galactose (growth?), one may want to know what is critical for this response, for instance is it the amount of a specific enzymes or set of enzymes? and how reading the pGal1-GFP activity reflects this response. Also, it would be useful to know what the effect of the destabilized GFP is on the inferences made. I understand that this is used to assess transcriptional output and not protein accumulation. However, if proteins of the pathway and long-lived, could we expect some of the features of the GFP response such as bi-modality to be irrelevant to the actual activity of the pathway itself? Could long-lived proteins attenuate this bi-modality and make it unimodal?

Thank you for this suggestion. We made additional simulations and we show in Supplementary Figure 12 that *i*) the binary-vs-gradual induction of pGal1-GFP also corresponded to binary-vs-gradual induction of Gal1p, Gal3p and Gal80p and *ii*) the reporter half-life (short for GFP_{pest} and longer for YFP) did not affect the response type. These results are now mentioned in the revised Supplementary Text 1.

• One major weakness I see from the study is that the conclusion is on the use of dynamical modeling to assign function to natural allele variation but the example chosen affects the binding of Gal3p to Gal80p. The analyses and some statements in the abstract for instance (line 38) give the impression that this stands out from the modeling directly and that the model could lead to this. However, the impression from the paper is that this could have been found simply by

looking at the protein complex structure and conservation of the residues, and that the dynamical modeling was not instrumental in identifying this mutation as critical for binding. The title also suggests that the function of the alleles that are altered can be discriminated from the model but the model uses two parameters K_{gal} and $pgal3$ that themselves can be tuned in many different ways by mutations and it is not even clear how these two parameters can be discriminated by the model. I guess this issue can be resolved by adjusting the scope of the paper so that it fits the actual finding. Another approach would be to create targeted GAL3 alleles using mutations of known or predicted effects to show that these parameters can be tuned (or not) independently and how one could for instance possibly dissect our all different parameters that go into the "strength" of Gal3p.

We agree. To test if the model can capture the known (or anticipated) effect of a mutation, we worked on three sets of experiments: i) introduction of specific point mutations on a plasmid carrying the full-length GAL3 gene that we ectopically integrated in a gal3null strain, ii) Tagging of GAL3 with an auxin-inducible degron system so that the degradation rate of Gal3p can be tuned experimentally and iii) Crispr/Cas9-introduction of a point mutation targeting the binding of Gal3p to ATP.

Regretfully, we experienced technical difficulties along strategies i) and ii).

Strategy iii) was, however, excitingly successful. We observed (as expected) that the W117 residue was crucial for Gal3p function. A W117A mutation caused a binary and weak response. Model-fitting showed that we could capture the expected effects on K_{gal} and ρ_{GAL3} , given that galactose binds first and ATP second in the course of Gal3p activation. These results were added to the revised text (lines 277-298, Supplementary Fig. 6) and the corresponding analysis of the model is explained in Supplementary Text 1 (section E-3).

Minor points

- Line 280: Does using an intermediate galactose concentration actually reflects a condition that is not used for parameter estimation?

It is correct that the specific condition of 0.2% concentration was not used for parameter estimation. We agree that it is an intermediate induction as compared to the ones used for estimation (between 0.1 and 0.5). Testing predictions of inter-strain differences outside this range is difficult because responses of the various strains become similar at inductions above 0.5%.

-References on the prediction of functional impacts of SNPs from line 66 to 69 appear to be from a few years ago (ref 2 for instance). It may be worth including more recent ones.

Besides sequencing/genotyping, recent methodological efforts have mostly been on finding more or « better » genes, or relevant gene modules (e.g. pubmed 27664809, 28243742, 28000566) but, as far as we know, not much on inferring SNP molecular functions. A deeper search picked the elegant recent study of Guo et al. 2016 who combined lncRNA eQTL maps with DnaseI hypersensitivity correlations to discover the regulatory function of a variant. The revised text now cites this work.

-Line 79 to 82. Since the work is put into the context of personalized medicine, i.e. using a patient's genotype and other information to predict, diagnose and treat diseases, I would be useful to briefly explain what we mean in practical terms by 'identifiable', sufficiently "constrained" and parameters that can be "reliably inferred". This is particularly important with respect to other approaches based on machine learning for instance in which we can make predictions without knowing how the system works, which is thought to be sufficient for many medical applications.

Yes, we have revised the text accordingly.

- Line 198. Many statements are made about the network as if they were universally true, for instance that Gal80p forms an homodimer etc. I suspect that this knowledge derives from studies performed on a single strain. In the light of the diversity uncovered in this study, would it be useful to specify this somewhere? In some ways, because of polymorphism, there is not "a" GAL pathway, there are many different ones.

Yes, absolutely. Because of historical developments of different laboratories, it's not a single strain but different (related) ones used as references (e.g. BY, W303...). The revised text now specifies that this knowledge « derives from reference laboratory strains ».

- Are parameter values used and reported in Table S1 derived from the same genetic background? If not, does it matter for this work?

Most of these values were taken from Hsu *et al.* Nat Comm. (2011) who used strains with the same reference genetic background (S288c) as our BY strain. This, however, probably does not matter regarding our conclusions because the relative ratios $K_{Gal}/K_{Gal}(BY)$ and $\rho_{Gal3}/\rho_{Gal3}(BY)$ only weakly depend on the precise values of the other parameters (see Fig. 4a).

- It would have been useful to know how the different GAL3 alleles affect the network dynamics when in their own genetic background, i.e. comparing the endogenous GAL3 allele in a background with that of the GAL3 BY alleles. It could have illustrated the contribution of the alleles versus other loci in differentiating the strains.

We have tested three natural backgrounds and the results are now reported in Supplementary Figure 10. After a preculture in raffinose (no glucose), these strains tended to form packs of cells, which made single-cell quantifications difficult. However, strain Y12 provided enough isolated cells for measurement, and its induction dynamics was much more binary than the dynamics of the BY strain carrying the $GAL3^{Y12}$ allele. This illustrates the importance of other loci and this conclusion is now mentioned in the revised discussion (lines 533-536).

- Figure 2. Is Gal4p not on the figure?

The revised legend now explains that it is not shown because its dynamics are not included in the model.

Reviewer #2:

The authors study how alleles of the GAL3 gene affect the induction kinetics of the GAL network and fit binding parameters to the observations. They observed two different types of responses with the different alleles: binary and graded. The model describing the above responses in different conditions should be worked out more precisely before publication:

I. Binary / graded responses.

IA. (minor) The terms should be used more consistently. For, example the authors write: "Inducibility increased with the concentration of galactose, with low concentrations causing a probabilistic induction (binary) and high concentrations a deterministic one (gradual)."

Even a graded induction can be stochastic. Therefore, the authors should use only phenomenological descriptions, i.e. binary and graded (or gradual), and omit terms that are used to describe model types (deterministic models vs probabilistic - stochastic models).

Yes. The revised text now consistently uses the terms binary/gradual (corrections in lines 118, 156, 243, 342).

IB. (major). Binary response is usually - but not always - a sign of an underlying deterministic bistability. For example, a binary response can arise in the absence of bistability [PMID: 22125482]. The authors should determine if the deterministic model corresponding to the stochastic model with the fitted parameters displays monostability or bistability (ie. the model consisting of the ordinary differential equation has two stable solutions). Then it will be possible to show if the binary and graded responses correspond to bistable and monostable parts of the parameter space. If the binary response is associated with a monostable solution, this may be a sign of a slow transient [PMID: 27498164]. This may be the case since the authors indicate for some alleles that: "Strains having a low decision threshold, such as GAL3YJM421, displayed a transient binary response, and strain GAL3BC187 had a high decision threshold and responded gradually.". The transient response can be often slower than usually expected. In fact, steady-state condition is rarely attained in realistic stochastic systems.

We thank the reviewer for raising this point: it is indeed important to distinguish "bistable" from "binary", which could be a transient regime towards monostable steady-state. We have added an analysis of the deterministic stability of the system, which is explained in revised Supplementary Text 1. The corresponding bifurcation diagrams were added to Figure 4 (panel d). They show that the "positions" of the strains in the parameter space correspond to transient binary inductions and not to bistability at steady-state. Convergence to steady-state may indeed be very slow (more than 10h, which is well beyond the typical time window used in experiments). This is now explained in the revised text (lines 237-239 and 343-347, and Supplementary Text 1).

2. Galactose / Glucose conditions: The expression of GAL4 depends on glucose (e.g. PMID: 915298, PMID: 28333434). However, the authors consider the GAL4 expression to be constant. Of course, the metaparameters can be fitted with constant GAL4 expression because the experiments were performed in galactose / raffinose. However, the glucose experiments (e.g. Fig. 5) require the inclusion of the GAL4 response to glucose to see if the K_{gal} and ρ_{Gal3} parameters maintain their relationship with respect to the glucose threshold. The authors may also show the GAL4 response by measuring GAL4 RNA.

We consider GAL4 expression to be constant during the dynamic induction experiments, which take place in the absence of glucose, not across the data of Fig. 5 which are at steady-state. The glucose/galactose decision threshold is used as a different trait of the strains and it is compared to the K_{Gal} and ρ_{Gal3} traits that we estimated independently in absence of glucose. This is now more clearly explained in the revised text (lines 366-368).

3. Consistent distinction between the direct and indirect fitting and observation.

The authors use a precise definition of the metaparameter (e.g. "Second, even in the simple context of our study, not all parameters of the model were identifiable and it was necessary to aggregate several of them into a meta-parameter".) to indicate that it is an "indirect parameter". However, they cite observations as if they were direct even if they are indirect. E.g. " that the dynamics of nucleocytoplasmic trafficking were too slow to explain the fast induction of transcription²⁴."

However, the binding and shuttling in those experiments are lumped and are not distinguished. The Gal80-GFP translocation to the cytoplasm depends both on the transport rate and the Gal4-Gal80 dissociation rate. Two latter could have been eliminated in Gal4 deletion strains. Thus, the above observation can be interpreted in terms of a metaparameter.

Therefore, I suggest a more precise formulation in such cases.

Yes. The revised text now cites Egriboz *et al.* 2011 as follows: "the slowness of the nucleocytoplasmic translocation of Gal80p, which depends both on transport rates and on the Gal4p:Gal80p dissociation rate, contrasts with the fast induction of transcription".

Thank you for sending us your revised study. We have now heard back from the referee who was asked to evaluate your study. As you will see below, this reviewer is satisfied with the modification made and thinks that the study is now suitable for publication.

Before we can formally accept your study for publication, we would ask you to address some editorial issues listed below.

REVIEWER REPORT

Reviewer #1:

The authors made the modifications I requested in a satisfactory manner.

Corresponding Author Name: Gaël YVERT
Journal Submitted to: Molecular Systems Biology
Manuscript Number: MSB-17-7803R